# Modelling genetic stability in engineered cell populations

Duncan Ingram [1] & Guy-Bart Stan [1] ✉

Predicting the evolution of engineered cell populations is a highly sought-after goal in biotechnology. While models of evolutionary dynamics are far from new, their application to synthetic systems is scarce where the vast combination of genetic parts and regulatory elements creates a unique challenge. To address this gap, we here-in present a framework that allows one to connect the DNA design of varied genetic devices with mutation spread in a growing cell population. Users can specify the functional parts of their system and the degree of mutation heterogeneity to explore, after which our model generates host-aware transition dynamics between different mutation phenotypes over time. We show how our framework can be used to generate insightful hypotheses across broad applications, from how a device's components can be tweaked to optimise long-term protein yield and genetic shelf life, to generating new design paradigms for gene regulatory networks that improve their functionality.

The engineering of cells with synthetic DNA is one of the most promising technologies of recent decades[1], facilitating applications from producing life-saving pharmaceuticals[2] to making biomaterials with revolutionary properties[3]. While significant progress has been made on the methods underlying these achievements, a number of problems have continued to prevent them from reaching their full potential. One recurring issue is the ubiquitous effect of evolution; when sustaining a population of cells engineered with synthetic DNA, mutations will typically disable that DNA's function and lead to its progressive removal from the system. While the effects of this are generally not critical in laboratory experiments, they often have significant consequences when enacting larger-scale projects, such as stunting the yield of synthetic protein from cells in bioreactors[4], causing biosafety concerns when deploying cells into the environment[5], and allowing devices to evolve in unpredictable ways[6]. Being able to understand, predict and control evolution, therefore, has the potential to transform cell engineering from an already-promising tool to a leading technology in modern society.

Our quantitative understanding of how mutations spread in populations has developed since Fisher[7], Haldane[8] and Wright[9] pioneered mathematical descriptions of allele frequency changes in systems. Different formulations of these models have arisen to account for other modelling techniques[10,11], more complex population structures[12], and the presence of external selection pressures[13], but they are all united in describing how genotypes change over time. Since the advent of these models, concepts like 'fitness' and 'fixation' have been used to enhance our description of mutation spread[14], enabling us to better interpret how systems adapt given different starting conditions. While such models capture changes on the scale of individual genes, applying them to evolution in modern-day synthetic systems is inherently more challenging, where accounting for variation amongst smaller genetic parts and intricate genetic constructs is necessary. Not only do different combinations of promoters, ribosome binding sites, coding sequences and terminators create large variations in the potential evolution of a system, but when genetic components are designed with regulatory feedback[15], the scope of mutation dynamics soars. As a result, successfully describing evolution in synthetic systems requires a more targeted approach that can consider the combinatorial impact of mutating multiple linked genetic parts.

Current progress in capturing the mutation dynamics of synthetic constructs is varied. Experimentally, researchers have developed a litany of techniques that constrain evolution, such as removing repeats and methylation sites[16], using different combinations of genetic parts[17], and coupling the expression of a synthetic device to an essential gene[18]. While these experiments shed light on the workings of specific mutation mechanisms, they provide little insight into how evolution can be predicted a priori. More general and quantifiable approaches have resulted using deterministic models that define separate parameters for (i) the genetic stability of a synthetic construct and (ii) the selection pressures that act on the cells containing the construct[19,20]. Here, genetic stability is captured by assigning a parameter for the rate at which a function-

[1]Centre of Excellence in Synthetic Biology and Department of Bioengineering, Imperial College London, London, United Kingdom.
✉e-mail: g.stan@imperial.ac.uk

disabling mutation occurs. A number of computational tools[21–23] exist to quantify this rate, typically focusing on how the construct's sequence composition influences errors during DNA replication[24]. Selection pressure, meanwhile, is modelled using a parameter for the cell's growth rate, noting that cells typically grow faster when not expressing additional genes[25]. The larger the difference in growth rate between cells that do and do not express synthetic material, the faster the mutant phenotype grows in the evolving population. While these models are good starting points, they are unable to capture the myriad of mutation dynamics seen in typical experimental data[17]. Recent discussions in the literature[6] have attempted to address this issue by suggesting that dynamic evolutionary landscapes can be used to represent an evolution in synthetic systems. Such landscapes would display the likely evolutionary routes of devices by adapting in response to sequence composition, fitness-based pressures, and functional effects. While enticing, the vast range of mutation effects means that implementing them would require overcoming significant computational challenges, and as such, no implementations have been made.

More tangible routes to improving models of mutation spread in synthetic systems could result by focusing on more targeted aspects of evolution. One such area of improvement would be linking the growth rate of cells to synthetic gene expression, such that modelling constructs with different designs naturally results in different cell growth rates and thus, different selection pressures. Implementing this would require capturing the effects of gene expression burden, such that expressing additional proteins reduces the availability of shared cellular resources and stunts cell growth[26]. These effects are especially important to account for under high expression levels as the resulting stresses are known to cause additional growth defects and unintended behaviours[25,27]. The prevalence of these phenomena in synthetic systems has since motivated the development of host-aware[28] cell models that consider how resources are shared between the host and synthetic constructs[29–34], although their application to population-scale evolution is scarce. Another area of improvement would be accounting for how individual mutations can have varied effects, akin to the 'distribution of effects of mutations' in population genetics[35,36]. This trait has been applied across many scales, from describing phenotypic variety in cancer cells[37] to quantifying epigenetic variation[38]; however, like the use of host-aware models, it has not yet been used to capture evolution in engineered populations. Doing this would require a more targeted approach, such as modelling which genetic parts are affected by mutations and the extent to which they are affected.

To address the gap in our ability to capture varied mutation dynamics in engineered cell populations, we here-in present a modelling framework that is mutation-aware, host-aware, and flexibly accounts for varied gene construct designs in line with present-day synthetic biology. The model allows a user to specify their device's genetic design and the degree of mutation heterogeneity to explore, after which it automatically generates equations that simulate the relevant fitness effects and selection pressures in the evolving population. Throughout the paper, we show how our framework can generate important hypotheses about the effects of evolution in real-world systems, which in the future could produce new insights into strategies for assessing and mitigating the effects of evolution in synthetic devices. For example, for systems focused on protein production, we suggest how a synthetic device's DNA design can be tweaked to optimise long-term protein yield and genetic shelf life. In addition, we show how our model can propose new mutation-driven design paradigms for gene regulatory networks and illustrate this using the toggle switch and the repressilator.

## Results

### A modular framework captures varied mutation dynamics
The accumulation of cells with mutated synthetic DNA can be modelled by considering transitions between different mutation phenotypes over time. A turbidostat is chosen as the growth setting such that the number of cells is kept at a constant value, $N$, by diluting with fresh media. In its simplest form, the model encompasses two states: engineered cells (E-cells) of quantity $E$, which have fully-functioning synthetic DNA, and mutant cells (M-cells) of quantity $M$, whose synthetic DNA has been mutated and rendered inactive (Fig. 1a). These cells grow at rates $\lambda_E$ and $\lambda_M$ respectively, and all cells belonging to a particular state are assumed to be identical. During the course of a cell cycle, a mutation that fully inactivates the synthetic DNA can occur with probability $z_M$, leading to the production of one E-cell and one M-cell upon division. By considering the growth rate of each cell type, the division rate of an E-cell into one E-cell and one M-cell is $\lambda_E \cdot z_M$, while the division rate into two E-cells is $\lambda_E \cdot (1 - z_M)$. Here, growth rate is analogous to fitness, such that cells with higher growth rate are fitter and are more likely to be selected for within a growing population. Mutations are assumed to be irreversible such that M-cell division always produces two M-cells at a rate of $\lambda_M$.

When the optical density (OD) in a turbidostat surpasses a target value, cells are typically diluted by a fixed volume of media which causes the OD to drop. This creates a periodic rise and fall of cell number around a fixed value over time. For simplicity, we approximate this process by assuming that dilution occurs instantaneously in response to cell growth, keeping the OD at the target value, and hence keeping the number of cells at a fixed value, $N$. This requires a dilution function 'dil', which, when $E + M > N$, removes cells in proportion to their abundance above $N$:

$$\text{dil} = \begin{cases} E + M - N, & \text{if } E + M > N \\ 0, & \text{otherwise} \end{cases} \tag{1}$$

Based on these considerations, an ordinary differential equation (ODE) model capturing the dynamics of each cell type can be formulated as follows:

$$\dot{E} = E \cdot \lambda_E \cdot (1 - z_M) - E \cdot \text{dil}, \tag{2}$$

$$\dot{M} = E \cdot \lambda_E \cdot z_M + M \cdot \lambda_M - M \cdot \text{dil}. \tag{3}$$

In order to simulate realistic cell behaviours, parameter values for growth rate should consider the impact of synthetic gene expression on host cell growth. We capture these effects by combining the state transition equations (Equations (2)–(3)) with the host-aware cell model from ref. 29, chosen due to its comparative simplicity and ease of manipulation compared to other host-aware models[30–32]. In its base form, it uses a set of ODEs that describes how essential resources are distributed in an engineered *E. coli* cell (Equations (S2–S14), where the prefix 'S' refers to content in the supplementary information), chosen due to its simplicity and ubiquity in cell engineering. In this model, the cell is assumed to contain a fixed protein quantity split between four fractions: 'ribosomal' for translation, 'enzymatic' for energy metabolism, 'heterologous' for synthetic gene expression, and 'housekeeping' which is auto-regulated to remain constant and constitutes the remaining protein mass. Protein expression is assumed to use a finite supply of cellular energy, and the growth rate is then calculated from the combined production rate of each fraction (Equation (S16)). As synthetic protein expression occurs, the cell's growth rate dynamically changes based on how many resources are diverted from growth-supporting processes (such as producing energy and ribosomes) to synthetic gene expression. For purposes of generality, this model focuses on the general effect of synthetic gene expression on cell fitness and does not consider the functional effect of the gene. For specific systems, however, additional functional effects could be added by modifying the equations that govern the growth-supporting processes in the cell, such as energy metabolism or ribosome production (Supplementary Note Section S4). While the model is structured based on *E. coli*, it is noted that it could be adapted to other bacterial hosts by changing the cell-based parameter values. For precise details of Weisse et al.'s framework, see Supplementary Note Section S1.1.

We integrate Weisse et al.'s framework into our mutation model by forming one set of ODEs for each cell state. A given set of ODEs contains

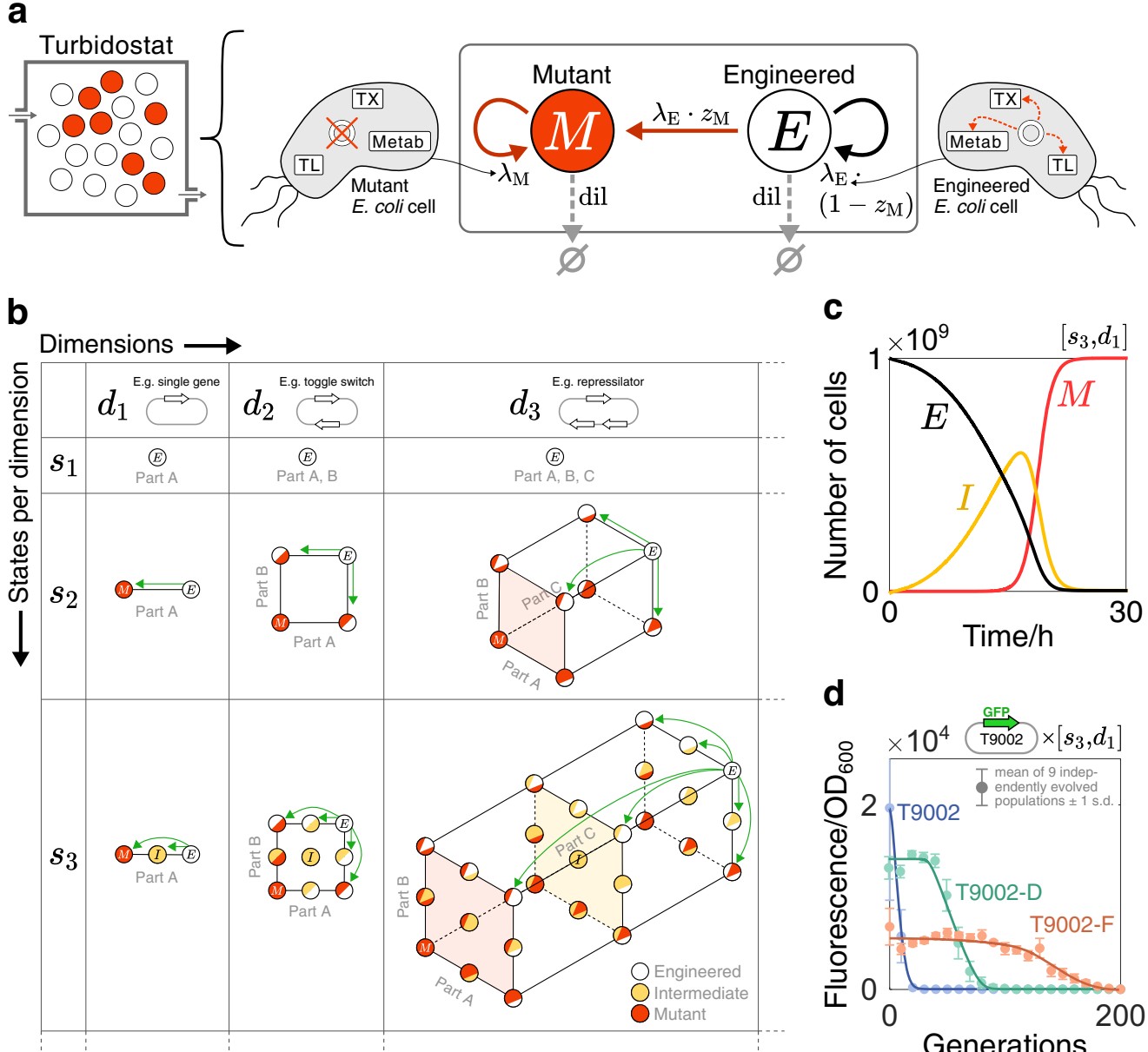

**Fig. 1 | The modular mutation-aware framework. a** The base model is represented by a two-state chemical reaction network in a turbidostat: engineered (E) cells express synthetic genes while mutant (M) cells do not. Cells belonging to a particular state are assumed to be identical. Transition rates are shown next to the arrows. $\lambda_E$, $\lambda_M$: growth rate of E-cells and M-cells, respectively; $z_M$: probability that an E-cell produces one E-cell and one M-cell upon division; dil: dilution rate from the turbidostat. Growth rates are inferred using a host-aware model of *E. coli* that considers transcription (TX), translation (TL) and energy metabolism (Metab). An independent cell model is used for each state to reflect differences in mutation phenotypes. **b** The two-state model is extended to consider variation in mutation location ('dimensions') and mutation severity ('states per dimension'). The dynamics of each mutable part are captured by states positioned along new orthogonal axes. Additional states added to a particular axis add variation in mutation severity, with states ordered in severity. For each state, a new cell model is

used that reflects the state's mutation phenotype. Transitions between states occur independently and are mono-directional such that mutation severity can only increase. *d*: number of dimensions. *s*: number of states per dimension. $[s_y, d_x]$ a framework where $[s, d] = [y, x]$. Some areas between states are shaded to help visual clarity. **c** Simulation of a $[s_3, d_1]$ framework where a single gene's promoter is considered to vary between three mutation states: E, I (intermediate) and M. The number of cells belonging to each state is plotted over time. Parameter values: $[\alpha_E, z_M, \alpha_I, z_I] = [10^4, 10^{-9}, 6 \times 10^3, 10^{-2}]$. **d** Simulation results from the $[s_3, d_1]$ framework are fit to fluorescence decay data from ref. [17]. Each data set represents a variant of the synthetic construct T9002 grown in *E. coli* cells. For simulations, fluorescence is the total amount of synthetic protein currently in the population and 'time' is converted to 'generations' using the cells' growth rates. See Supplementary Note Section S2 for parameter values and fit statistics.

an equation for that state's cell quantity (such as Equation (2) for E-cells) alongside the host-aware cell equations (denoted as a subset of equations called 'Ψ', Supplementary Note Section S1.1) from[29]. Within this structure, each equation set is adjusted to reflect its associated state's mutation phenotype by considering specific parameter values for the synthetic construct's expression dynamics. One such parameter is the E-cell's

maximum transcription rate ($\alpha_E$) which can be interpreted as the construct's promoter strength. By setting $\alpha_E$ to 0, for example, transcription is fully inhibited and the synthetic gene's expression becomes inactivated, hence capturing the behaviour of an M-cell. To model the dynamics of the whole population, each set of ODEs for all cell states are simulated together. By letting $\dot{\Psi}_X$ denote the host-aware ODEs specific to

state X, the full set of ODEs composing our two-state mutation-aware model is defined as:

$$\text{Two-state mutation-aware model} \triangleq \left\{ \dot{E}, \dot{\Psi}_E, \dot{M}, \dot{\Psi}_M \right\}. \tag{4}$$

In reality, mutations to synthetic constructs can affect a cell's phenotype in different ways depending on (i) the functional part they affect (mutation location) and (ii) the degree to which they affect it by (mutation severity), otherwise known as the 'fitness effect'. To capture these aspects of mutation heterogeneity, the two-state framework can be augmented by considering additional states that represent different mutation phenotypes (Fig. 1b). Location-specific mutations can be modelled by adding a dimension to the framework in Fig. 1a for each new part being considered. We define a 'dimension' as a distinct functional unit, such as a promoter or an RBS. Fine-graining mutable elements into further sub-functions could be considered, such as splitting a promoter into −35, spacer and −10 regions; however, these details would need to be captured in the corresponding cell model equations. In order to analyse the mutation dynamics of $x$ parts simultaneously, $x$ orthogonal axes are drawn whereby the states along one axis correspond to one part's mutation state. As such, moving parallel to an axis represents a change to the mutation phenotype of that axis' associated part. We define the parameter $d$ as the number of dimensions in a framework, with the notation $d_x$ denoting that $d = x$. For example, column '$d_2$' in Fig. 1b represents frameworks that model mutations into two parts. As depicted in Fig. 1b, dimensions can also represent mutations to parts across multi-gene constructs: $d_3$, for example, could capture the mutation dynamics of a three-gene repressilator by considering each gene's promoter for mutation. We assume that mutations to different parts develop independently, such that transitions are restricted to one axis at a time. Furthermore, we assume that each part functions independently, such that the occurrence of one mutation does not influence the probability of another.

Variations in mutation severity can be included via additional states along each axis. Cells within these intermediate (I) states are modelled using parameter values that change the activity of the corresponding part, but do not completely inactivate it. For a gene's promoter, for example, the maximum transcription rate of cells within an I-state ($\alpha_I$) would be distinct from $\alpha_E$. We focus our analyses on mutations that decrease the activity of a part, such that $\alpha_E > \alpha_I > 0$; however, this is not a requirement and the implications of activity-increasing mutations are discussed in Supplementary Note Section S4. Mutations with different severities are assumed to occur with distinct probabilities, requiring different parameters for the probabilities of creating I-cells ($z_I$) and M-cells ($z_M$) upon division. Additional I-states can be added to each axis to represent states with varying degrees of mutation severity, with these states being ordered by decreasing part functionality. Furthermore, we allow parts to sustain mutations of any severity, meaning that individual transitions can bypass I-states. We assume that mutations are irreversible, however, meaning that previous states cannot be revisited. Akin to the parameter $d$, we define the parameter $s$ as the number of states per dimension, with the notation $s_y$ denoting that $s = y$. For convenience, we use the notation $[s_y, d_x]$ to represent a framework with $x$ dimensions and $y$ states per dimension. Frameworks up to $[s_3, d_3]$ are shown in Fig. 1b; however, our model generates the appropriate framework structure for any integer values of $s$ and $d$.

When using frameworks with multiple mutation states, care needs to be taken when interpreting the core mutation parameters. First, an association between the maximum transcription rate, $\alpha_E$, and a construct's promoter strength can be obtained by measuring what parameter value corresponds to the maximum protein expression level of a single construct. For the probability of a severe mutation, $z_M$, values can be inferred using tools such as the EFM calculator; however, care should be taken regarding the proper interpretation, as outlined in Supplementary Note Section S2. For frameworks with $s > 2$, it is inherently challenging to give meaning to the parameters that govern specific intermediate mutation pathways, such as E-to-I. In reality, cell populations likely undergo multiple mutation pathways with subtly different

effects, so values of $\alpha_I$ and $z_I$ should not be interpreted as describing one specific mutation pathway. Instead, it is more appropriate to interpret the combined effect of these parameters as the 'degree of mutation heterogeneity' in a system, which can be loosely defined as the extent to which mutation dynamics deviate from the foundational E-to-M mutation pathway. In turn, systems with high mutational variation will be best modelled by combinations of $\alpha_I$ and $z_I$ that result in sustained I-subpopulations. If a user feels that their experimental dynamics are not sufficiently captured by an $s_3$ framework, they can then explore frameworks with $s > 3$.

A complete set of equations describing any framework can be represented by generalising Equation (4). By denoting a general mutation state as $X_i$, where $i \in [1, n]$, the full set of ODEs composing our mutation-aware model is defined as:

$$\text{General mutation-aware model} \triangleq \left\{ \dot{X}_1, \dot{\Psi}_{X_1}, \ldots \dot{X}_n, \dot{\Psi}_{X_n} \right\}. \tag{5}$$

As before, $\dot{X}_i$ represents the equation for the cell quantity in state $X_i$, and $\dot{\Psi}_{X_i}$ represents the set of host-aware cell equations in state $X_i$ (Section S1.1). Within these equations, the parameters governing the synthetic construct's expression, such as the maximum transcription rate $\alpha_{X_i}$, are chosen to reflect the mutation phenotype of the cells in that state.

To illustrate typical changes between states over time, a simulation of the $[s_3, d_1]$ framework is shown in Fig. 1c where a single gene's promoter is considered to vary between three mutation states. Here, all cells start in the E-state (black line), before transitioning to I- and M-states (yellow and red lines, respectively). In a two-state population model, the E-state and M-state curves would be expected to have symmetric rates of change, as any cell leaving the E-state becomes part of the M-state by definition. The presence of an I-state, however, skews these curves, as is seen by the E-state curve having an inflection point below its half-maximal value. This added asymmetry from partially-inactivating mutations may be important when capturing a larger variety of mutation dynamics in cell populations.

To demonstrate the benefits of capturing mutation heterogeneity with our model, we use our $[s_3, d_1]$ framework to fit simulation results to experimental data from ref. 17 (Fig. 1d). As in Fig. 1c, we apply this framework by modelling a single gene's promoter varying between three mutation states in order to approximate the effect of mutations that fully inactivate the construct. In their experiments, the authors engineer *E. coli* with variants of the same fluorescent-tagged synthetic construct (T9002 and six variants appended '-A' to '-F'), and record how each population's fluorescence decays over time due to the onset of mutations. Fits to three of these experiments are shown here (T9002 in blue, T9002-D in green and T9002-F in orange), with more complete details of fitting to all experiments given in Section S2. In our simulations, 'fluorescence' represents the total amount of synthetic protein currently in the population, and 'time' is converted to 'generations' using the cells' growth rates. The full details of these adjustments are given in Supplementary Note Section S2.

The different designs associated with each T9002 construct cause changes to both the mutation probability of the construct and the associated growth rate of the cells. The precise combination of these factors is typically unknown without extensive experimental investigation; however, they can be estimated by fitting key mutation parameters in our model ($\alpha_E, z_M, \alpha_I, z_I$) to the experimental data. For the experiments shown in Fig. 1d, optimal fits are obtained for T9002 and T9002-F using values for $\alpha_I$ and $z_I$, while fitting to T9002-D was best achieved without these parameters. Given that good fits can be obtained in some instances without describing intermediate mutations, it is evident that our model strikes a balance between the number of parameters and the ability to capture complex behaviours. While this is not equivalent to obtaining a complete predictive association between a system's genetic design and its evolutionary dynamics, experimental interpretations can still be obtained for our framework's key mutation parameters, as outlined previously. A more thorough discussion of these fits and their underlying methodology is given in Supplementary Note Section S2.

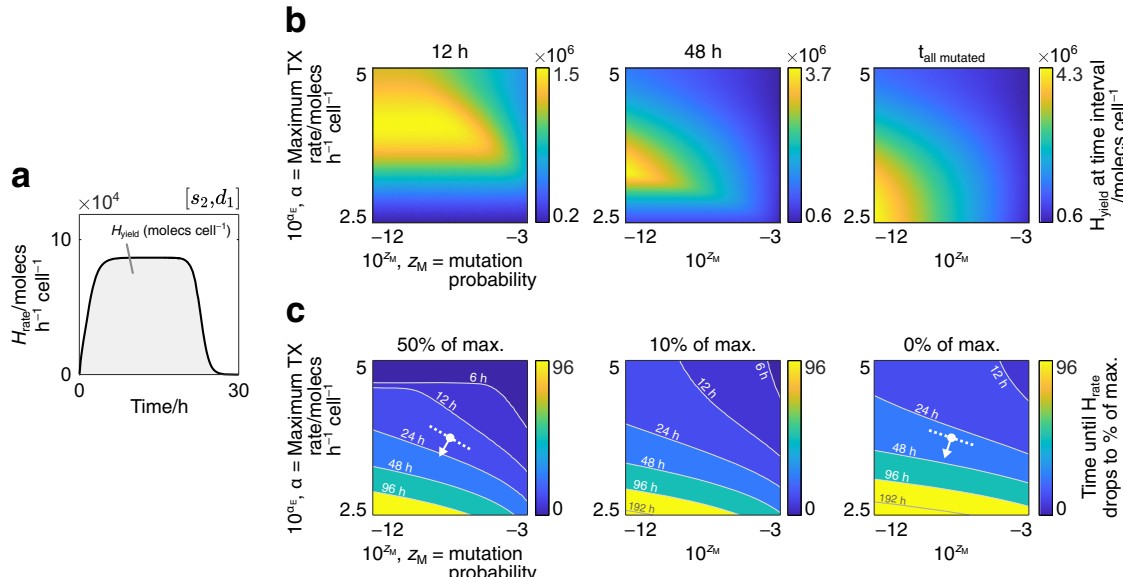

**Fig. 2 | Modelling protein yield and viability of protein production. a** Simulation of a framework where fully-inactivating mutations of a single gene's promoter ($[s_2, d_1]$) are modelled. The rate of synthetic protein production, $H_{rate}(t)$, is plotted. The synthetic protein yield, $H_{yield}(T_0, T)$, is the area under this curve. Parameter values for the synthetic gene: $[\alpha_E, z_M] = [1.05 \times 10^4 \text{ moles h}^{-1}\text{cell}, 10^{-12}]$. **b** Heat maps of synthetic protein yield for the $[s_2, d_1]$ framework in (**a**), calculated by varying the mutation probability ($z_M$) and the maximum transcription rate ($\alpha_E$). Yield measurements are taken at three intervals ($[0, T]$ where $T \in \{12, 48 \text{ h}, t_{\text{all mutated}}\}$) to show

how the optimal parameter combinations change over time. **c** Contour maps of the time taken for $H_{rate}(t)$ to drop to certain percentages (50%, 10% and 0%) of its maximum value for the $[s_2, d_1]$ framework in (**a**), calculated by varying $z_M$ and $\alpha_E$ as in (**b**). Tracing any contour line gives the combinations of $z_M$ and $\alpha_E$ that ensure a minimum protein production rate (given by the map) for the duration of that contour's time value. White dots represent constructs with the same value of $\alpha_E$ and $z_M$, with arrows pointing in the direction of the maximum increase in contour time values.

## Synthetic construct design impacts protein production over cell generations

The impact of mutations on protein production is of particular interest in biotechnology, affecting both product yields and the reliability of experiments[4,39]. While cell strains with higher expression loads produce higher synthetic protein yields per unit time, they are also selected against faster in a growing cell population[17]. This trade-off between genetic shelf life and protein production yield is often unclear, and can be explored using our mutation-aware framework.

In order to evaluate protein production through the lens of mutations, quantitative metrics can be defined in the context of our model. In a turbidostat, the yield of a given protein type can be defined as the cumulative amount of that protein extracted from cells that leave the chamber as a result of dilution. Here-in, 'heterologous' (H) will be used to describe proteins and variables related to the expression of synthetic gene constructs. The rate of H-production attributed to cells in a general state '$X_i$' can be calculated by multiplying the amount of H-protein in one of those cells, $H_{X_i}(t)$, the growth rate of one of those cells, $\lambda_{X_i}(t)$, and the abundance of that cell type in the turbidostat, $X_i(t)$. The rate of H-production in the entire population is therefore the sum over all cell types. To compare results between continuous cultures of different sizes, we can furthermore represent this quantity as a per-cell average by dividing by $N$. It follows that the per-cell average rate of H-protein production in the population (unit: moles $h^{-1}$ cell$^{-1}$) is:

$$H_{rate}(t) = \sum_{i=1}^{n} \frac{H_{X_i}(t) \cdot \lambda_{X_i}(t) \cdot X_i(t)}{N \cdot \ln(2)}, \quad (6)$$

with the factor $\ln(2)$ required to link growth rate to doubling time. In turn, the per-cell average protein yield (unit: molecs cell$^{-1}$) between two time points $T_0$ and $T$ is the integral of $H_{rate}(t)$:

$$H_{yield}(T_0, T) = \int_{T_0}^{T} H_{rate}(t) \, dt. \quad (7)$$

Intuitively, Equation (7) represents the amount of new cell mass that is converted to H-protein as the population grows during the time interval $[T_0, T]$.

Figure 2a shows how $H_{yield}(T_0, T)$ can be obtained from our framework in a simple case. Here we consider a population of cells expressing a single synthetic construct whose promoter is subject to fully-inactivating mutations ($[s_2, d_1]$). The promoter is chosen because its inactivation blocks all downstream gene expression, thereby approximating the effect of fully inactivating the gene. As $t$ approaches 30 h in this simulation, the accumulation of non-producing M-cells leads to $H_{rate}(t)$ decreasing to zero, meaning that no additional protein yield can be obtained.

When growing a cell population for synthetic protein extraction, it may not be desirable to wait until the mutations have become widespread before altering or terminating the experiment. This is because (i) beyond a certain time, the quantity of experimental resources required may outweigh the gains in protein yield and (ii) researchers may want to maintain a population above a particular threshold of active synthetic gene expression by restocking the turbidostat chamber with viable cells. For these reasons, it is useful to simulate the impact of different construct designs on protein production when measuring (i) the yield at fixed time intervals and (ii) the time taken for the population's protein expression rate to drop below a particular threshold. These can be calculated by varying parameters that are intimately tied to the construct's design: the mutation probability ($z_M$), which is significantly influenced by a construct's sequence composition[22], and the maximum transcription rate ($\alpha_E$), which depends on the choice of promoter and affects the cell's growth rate via the diversion of shared cellular resources towards heterologous gene expression rather than biomass production.

While it is intuitive that increasing $z_M$ decreases protein yield, the effect of increasing $\alpha_E$ is less obvious as higher expression rates typically correlate with slower growth rates. To explore this further, we calculate $H_{yield}(0, T)$ for the simple $[s_2, d_1]$ framework in Fig. 2a over different

intervals ($T \in \{12, 48 \text{ h}, t_{\text{all mutated}}\}$), and observe how the optimum combinations of $z_M$ and $\alpha_E$ are predicted to vary (Fig. 2b). Here, $t_{\text{all mutated}}$ is the time at which all cells in the turbidostat have become M-cells. Results are displayed over time as successive heat maps, with lighter regions indicating combinations of $z_M$ and $\alpha_E$ that produce more H-protein. As time progresses, the optimum value for $\alpha_E$ drops, suggesting that lower protein expression delivers higher protein yield when culturing cells over longer time periods. This can be explained via the trade-off between the rate of protein expression and the effect of selection pressure: while higher transcription rates deliver more H-protein per unit time, these cells use more shared cellular resources and so grow more slowly compared to those with weaker H-protein expression. They are therefore diluted from the turbidostat at a faster rate, leaving fewer protein-producing cells in the population. This adverse effect on protein yield is relatively small at the beginning of an experiment but becomes more significant as time progresses, leading to a steady decline in the optimal value of $\alpha_E$.

For the mutation probability $z_M$, the range of values that maximise protein yield becomes smaller over time, with later time points more significantly penalising higher values of $z_M$. This can be explained by considering that the accumulation of mutant cell types in a population accelerates over time. During early population growth, a limited number of mutant cells are produced according to the mutation probability, however, as these cells grow faster than other cell types, a tipping point is eventually reached that allows their combined growth to accelerate beyond the growth of other cell types. This aligns with the observed protein production dynamics from Figs. 1d, 2a, where a fast decline in protein production is typically seen following a sustained period of protein expression. Together, these results suggest that optimising a synthetic construct's sequence for mutation becomes more crucial the longer that protein production is sustained for.

A separate consideration is monitoring a population's rate of protein production over time. This may allow researchers to gain insight into their device's genetic shelf life and, in turn, allows them to predict when a system should be replenished with unmutated (E-state) cells. The impact of synthetic construct design on these aspects can be analysed using contour maps of the time taken for the population's protein expression rate, $H_{\text{rate}}(t)$, to drop to certain percentages of its maximum value, $\max(H_{\text{rate}}(t))$ (Fig. 2c). In each map, tracing any contour line gives the combinations of $z_M$ and $\alpha_E$ that ensure a minimum protein production rate (either 50, 10 or 0%) for the duration of that contour's time value. For example, if a user wants to design their construct to sustain a minimum of 50% production rate for at least 24 h, they could choose any parameter values bounded below the 24 h contour line in the left-most map.

In each contour map, it can be seen that higher $z_M$ and $\alpha_E$ lead to faster mutation accumulation, as displayed by the upper-right regions of the maps displaying contour lines with lower time values. This is as expected, as both of these parameters are positively associated with selection pressure. Other trends can be seen by comparing the efficacy of $z_M$ and $\alpha_E$ on increasing shelf life. To illustrate this, white dots are added to the '50%' and '0%' maps that represent a synthetic construct with equivalent maximum transcription rate and mutation probability. The arrows leaving these dots point in the direction of maximum increase in contour time values, such that changing $\alpha_E$ and $z_M$ in this direction is the most direct way to increase the genetic shelf life. As more mutations are allowed to accumulate in a population (moving between maps left to right), the direction of maximum increase becomes more vertical, meaning that $\alpha_E$ has an increasingly large effect on shelf life relative to $z_M$. These results suggest that, if researchers are committed to growing cells until cell viability drops to low values, they can better sustain synthetic gene expression by varying the cell's rate of protein production (e.g. by changing the promoter) rather than its mutation probability (e.g. by removing mutagenic sequence features). While this trend is specific to the construct design indicated by the red mark, it can be seen to hold for many other designs as the contour lines

straighten between maps (left to right) across large regions of the parameter space.

## Genetic toggle switches lose switching capacity over time

So far, we have shown how our framework can be used to explore the mutation dynamics of a single-gene construct with a simple protein output. Although this is useful to capture fundamental trends in synthetic protein expression, it does not represent more complex genetic devices such as those with multiple genes and regulatory components. To better demonstrate our framework's potential, we apply it to two of the most influential gene regulatory motifs in synthetic biology: the toggle switch and the repressilator. The mutation dynamics and subsequent protein expression behaviours of these devices are unintuitive a priori and so constitute prime examples for the application of our mutation-aware modelling framework.

In its simplest form, a toggle switch comprises two genes whose protein outputs mutually inhibit the transcription of the other. They are typically designed to be bistable, with the two asymptotically stable steady states corresponding to one gene being highly expressed while the other is silenced and vice versa. Specific external protein inhibitors can be added at a sufficiently high concentration to switch the toggle from one steady state to another. This capacity for switching is arguably the most fundamental feature of a toggle switch and is hence used in multiple applications, from the regulation of gene expression[40,41] to controlling biofilm formation[42]. Until now, theoretical analyses of the switching behaviour of toggle switches have assumed unchanging protein expression dynamics[43–45], whereas in reality, one would expect mutation accumulation to reduce the ability of these genetic constructs to switch between states.

In our analysis, we consider a symmetric toggle switch that expresses mutually repressing species protein-A and protein-B at concentrations $A$ and $B$, respectively (Fig. 3a, top). Our simulations start in the asymptotically stable steady state corresponding to the high expression of protein-A. At a given time, a species $I_A$ is added at concentration $I_A$ to inhibit the ability of protein-A to repress the expression of protein-B. The inhibitor is assumed to be evenly mixed in the population such that it is distributed to each cell type in proportion to its abundance. If $I_A$ is sufficiently high, a switch from the high protein-A stable state to the high protein-B stable state occurs in the cells that still express synthetic proteins. As time passes, however, the accumulation of mutations means that an equivalent concentration of inhibitor may lead to fewer cells switching from more protein-A to more protein-B. In other words, the onset of mutations may cause the population to lose its capacity to switch from one stable state to the other over time. These effects are qualitatively illustrated in the middle row of Fig. 3, which tracks the concentrations of $A$ and $B$ averaged per cell. If $P_{X_i}$ denotes the concentration of a protein within cell state $X_i$, and if the quantity of that cell type in the population is $X_i$, then the average concentration of that protein per cell ($P$, unit: moles cell$^{-1}$) is defined as:

$$P = \sum_{i=1}^{n} \frac{P_{X_i} \cdot X_i}{N}. \tag{8}$$

To quantitatively track a population's change in switching capacity, we add the inhibitor at successive time points and different concentrations, and record the number of cells whose protein concentrations satisfy $B > A$. When adding an inhibitor to any system, we assume that it is evenly mixed in the system and taken up by cells instantaneously. To compare the effect of the inhibitor across different simulations, at each time point that we add it, we record its concentration required to induce at least 50% of the population's cells to switch from one steady state to another. This metric does not consider the distance between fixed points in phase space, and as such, our measure of a population's switching capacity is defined purely as the ability to switch states rather than the strength of the corresponding switch.

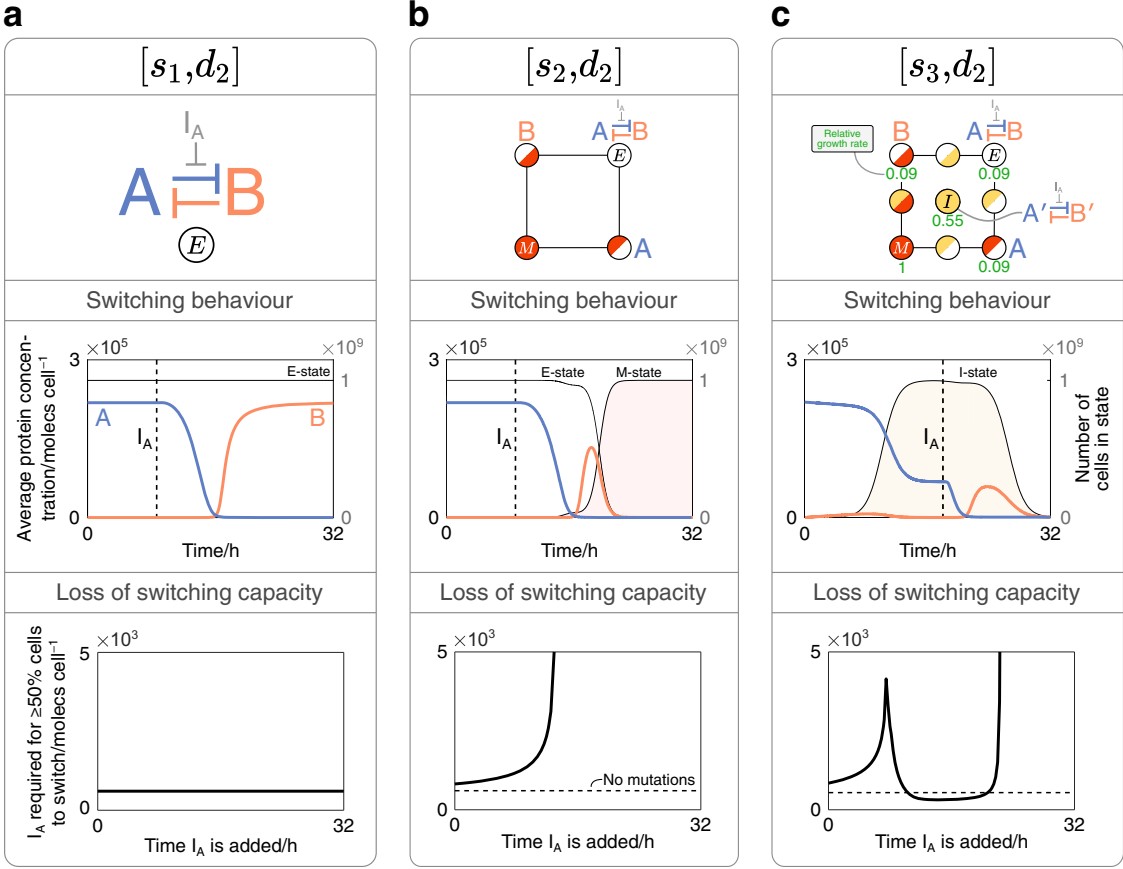

**Fig. 3 | Applying our framework to study the mutation-driven dynamics of a toggle switch. a** Simulation of a cell population expressing a toggle switch without mutations ([$s_2, d_1$]). *Top*: the gene network topology of a toggle switch whose genes express species protein-A and protein-B, and an inhibitor to the repression of protein-B labelled $I_A$. *Middle*: average concentrations of protein-A and protein-B per cell over time (left y-axis), and the number of cells in various states over time, as labelled (right y-axis). The inhibitor is added at $t = 9$ h (vertical dashed line) at a concentration of 1500 moles cell$^{-1}$, causing a transition from the high protein-A steady state to the high protein-B steady state. *Bottom*: the concentration of inhibitor required for at least 50% of the cells to transition from more-A to more-B. Variation in this quantity is monitored when adding inhibitors at different time points. For each synthetic gene, $\alpha_E = 10^5$ molecs h$^{-1}$ cell$^{-1}$. **b** A simulation of a population expressing a toggle switch where fully-inactivating promoter

mutations to both genes ([$s_2, d_2$]) are modelled. *Top*: each mutation state can be mapped onto the framework shown. All cells start in the non-mutated E-state (top-right). *Middle*: as in (**a**). With multiple cell types, the inhibitor is assumed to be distributed proportionally between each type. *Bottom*: as in (**a**), with a dashed line showing the concentration of inhibitor required when no mutations occur. For each gene, [$\alpha_E, z_M$] = [$10^5$ molecs h$^{-1}$ cell$^{-1}$, $10^{-6}$]. **c** An equivalent system to (**b**) but additionally considering partially-inactivating mutations to each gene's promoter ([$s_3, d_2$]). *Top*: as in (**b**), with partially-mutated genes annotated with a prime (') symbol. Numbers denote the growth rate of cells within select states relative to M-cells, in simulations without inhibitors. *Middle*: as in (**a**) and (**b**), but with the inhibitor added at $t = 18$ h. *Bottom*: as in (**b**). For each synthetic gene, [$\alpha_E, z_M, \alpha_I, z_I$] = [$10^5$ moles h$^{-1}$ cell, $10^{-6}$, $5 \times 10^3$ moles h$^{-1}$ cell$^{-1}$, $5 \times 10^{-2}$].

With no mutations (Fig. 3a), adding the inhibitor at a sufficiently high concentration will cause all cells in the population to switch from high protein-A to high protein-B, such as the dynamics shown in the middle panel. Furthermore, the constant value of $I_A$ in the bottom panel means that this switching effect is independent of when the inhibitor is added. This suggests that the switching capacity of the system does not change over time, a feature found in traditional theoretical analyses of toggle switches.

To explore the impact of mutations, we can first apply the [$s_2, d_2$] framework, which models fully-inactivating promoter mutations to each synthetic gene (Fig. 3b). The top panel shows how the states can be interpreted by adding the construct design that corresponds to the relevant mutation phenotype. When the inhibitor is added to this system at the same concentration and at the same time point as in the case with no mutations, different dynamics are seen: the effect of inhibition starts a transition from protein-A to protein-B; however, due to the accumulation of mutations, the average concentration of protein-B falls before any steady state value is reached. It can be imaged how the greater the onset of mutations in a system, the lower the concentration that protein-B reaches, and in turn, the fewer the cells satisfy $B > A$. In these cases, for an

equivalent number of cells to switch from more protein-A to more protein-B, a greater concentration of inhibitor would therefore be required. This relationship is shown in the bottom panel, which plots the concentration of inhibitor required to cause ≥50% of cells to switch from more protein-A to more protein-B when added at different time points. As shown, adding inhibitors at progressively later time points requires higher values of $I_A$, indicative of the system losing its switching capacity over time. This continues until the curve reaches a vertical asymptote, at which point the mutation accumulation is too great for switching to occur.

As explored with our varied framework designs, mutations may not completely inactivate a component, but could also partially reduce its activity. Such details may be important to capture when attempting to fully understand the dynamics of complex gene regulatory networks. These effects can be considered for the toggle switch by applying our [$s_3, d_2$] framework, which additionally models mutations that partially inactivate each gene's promoter (Fig. 3c, top). The construct designs associated with select states are shown, with genes containing partially-mutated components being annotated with a prime (') symbol. These partially-inactivating mutations cause genes to produce proteins at a lower rate,

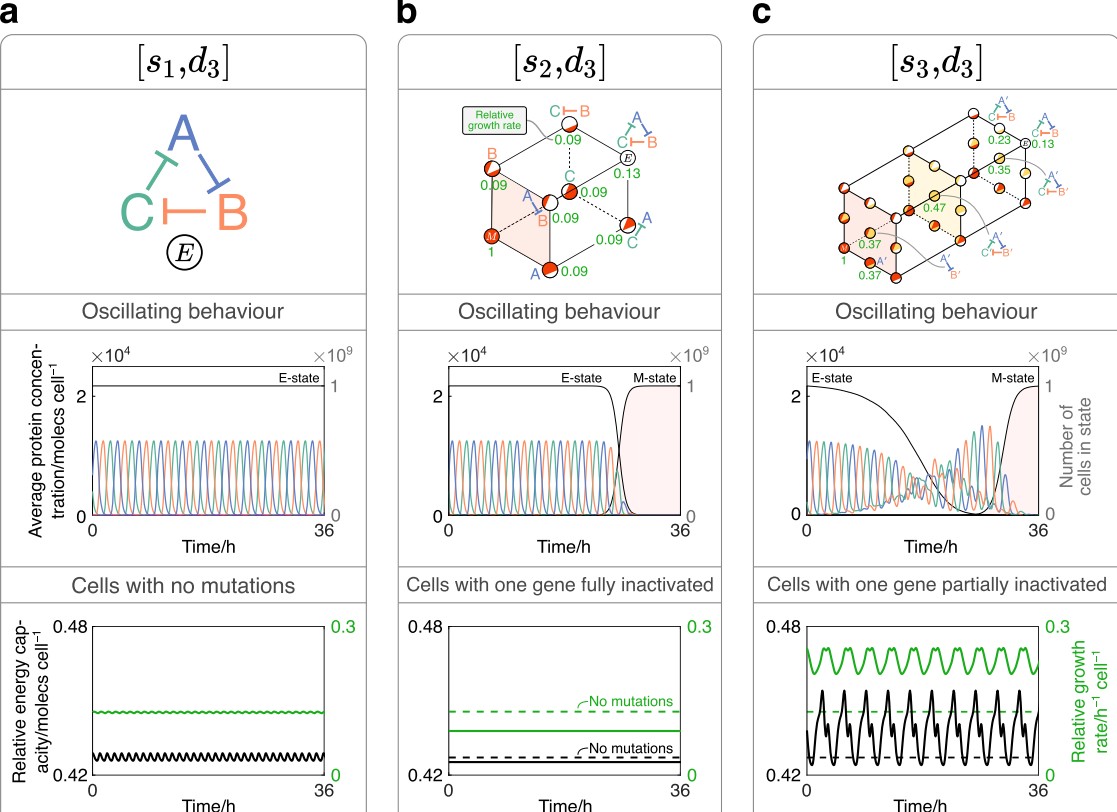

**Fig. 4 | Applying our framework to study the mutation-driven dynamics of a repressilator. a** Simulation of a cell population expressing a repressilator without mutations ($[s_1, d_3]$). Top: the gene network topology of a repressilator whose genes express species protein-A, protein-B and protein-C. Middle: average concentrations of protein-A, protein-B and protein-C per cell over time (left y-axis), and the number of cells in various states over time, as labelled (right y-axis). Bottom: energy capacity per cell relative to M-cells (left y-axis, black) and growth rate per cell relative to M-cells (right y-axis, green). For each synthetic gene, $\alpha_E = 10^5$molecs h$^{-1}$ cell$^{-1}$.
**b** Simulation of a cell population expressing the genes of a repressilator when fully-inactivating promoter mutations ($[s_2, d_3]$) are considered. Top: each mutation state can be mapped onto the framework shown. All cells start in the non-mutated E-state

(top-right). Numbers denote the time-averaged growth rate of cells within a state relative to M-cells. Middle: as in (**a**). Bottom: as in (**a**) but for cells with one synthetic promoter fully inactivated. Horizontal dashed lines are added to indicate the time-averaged energy capacity and growth rate for cells with no mutations. For each synthetic gene: $[\alpha_E, z_M] = [10^5$ moles h$^{-1}$ cell$^{-1}$, $10^{-6}]$. **c** An equivalent system to (**b**), but additionally considering partially-inactivating mutations to each gene's promoter ($[s_3, d_3]$). Top: as in (**b**) with partially-mutated genes annotated with a prime (′) symbol. Middle: as in (**b**). Bottom: as in (**b**) except modelling cells where one gene has a partially-inactivated promoter. For each synthetic gene, $[\alpha_E, z_M, \alpha_I, z_I] = [10^5$molecs h$^{-1}$ cell$^{-1}$, $10^{-6}$, $1.2 \times 10^4$moles h$^{-1}$ cell$^{-1}$, $1.2 \times 10^{-2}]$.

leading to reduced resource consumption and faster cell growth. This is indicated by the values adjacent to select states which show the growth rate of cells in those states relative to M-cells, in simulations without inhibitors. Cells with higher growth rates are more strongly selected for in a growing cell population, and as a result, their impact on the population's protein expression can often be seen. This effect is shown in the middle panel, where the number of cells with partially-inactivating mutations to both synthetic proteins is plotted over time (yellow region). Despite expressing synthetic protein at a lower rate, the accumulation of this cell phenotype enables the average concentration of protein-A to restabilise at a new lower value after initially declining, indicative of transitions from the E-state to the I-state.

Given that lower concentrations of protein-A inevitably require less inhibitor to cause a transition from more protein-A to more protein-B, the presence of cell phenotypes with higher growth can have interesting implications on a system's switching capacity. For example, when inputting inhibitor at $t = 18$ h after protein-A has dropped (vertical dashed line in middle panel), it can be imagined how the concentration required to cause ≥50% of cells to switch is lower than when no mutations are present. This can be seen in the lower panel of Fig. 3c, which shows that, while the population's switching capacity initially declines, it begins to regenerate at $t \approx 8$ h and results in a time period where the system's switching capacity surpasses that of a system with no mutations, as indicated by the solid line dipping below the horizontal dashed

line. After this, mutations to the faster-growing cell phenotype eventually lead to the system's complete loss of switching capacity, and the concentration of inhibitor required reaches a vertical asymptote, as with the $[s_2, d_2]$ framework. These results suggest that, while mutations inevitably lead to the complete loss of switching capacity, the presence of mutation heterogeneity can permit toggle switches to enhance their switching abilities, as measured by the number of cells that are able to transition from having more of one protein species to more of another.

## Repressilators are resistant to single points of failure

Repressilators are genetic oscillators typically consisting of three genes inhibiting each other in a one-way ring structure (Fig. 4a) with widespread application in synthetic biology[46]. Naturally-occurring genetic oscillators are robust to shifts in period or amplitude, often due to in-built feedback mechanisms[47,48]; however, when implemented synthetically, their ability to sustain regularly-repeating oscillations diminishes[49,50]. One cause of this loss of regularity may be the oscillator's genes losing functionality over time; however, experimental data that explores this is scarce. We, therefore, apply our framework to examine a synthetic repressilator's dynamics in a growing cell population, and in turn, explore how mutations may impact the regularity of oscillations.

In our analysis, we consider a repressilator that expresses species protein-A, protein-B and protein-C at concentrations $A$, $B$ and $C$,

respectively. To understand the construct's behaviour at the population level, we plot the average concentration per cell (Equation (8)) of each protein and characterise the oscillations that result. To further understand how the repressilator's activity impacts the host, we additionally record the energy capacity of cells in certain states relative to that in the M-state (Equation (S2)). In the host-aware cell equations, energy is essential for the expression of all genes and thus constitutes a key growth-limiting resource. As with the protein species, this variable is recorded as the average concentration of molecules per cell.

With no mutations (Fig. 4a), the construct behaves as expected, with the proteins oscillating with uniform period and amplitude (middle panel). The energy capacity and growth rate of each cell is recorded over time, shown in the bottom panel as black and green lines corresponding to the left and right y-axes, respectively.

As with the toggle switch analysis, we can first apply a framework that models fully-inactivating mutations to each synthetic gene's promoter (Fig. 4b). The top panel shows how the construct design that corresponds to each mutation phenotype can be mapped to each state from the $[s_2, d_3]$ framework. Values adjacent to the states indicate the time-averaged growth rate of cells within that state relative to that of M-cells. When recording the protein concentrations over time, we see that the oscillations remain robust for an extended period of time before their amplitude eventually reduces to zero (middle panel). This is also reflected by the number of cells in the E- and M-states (right y-axis, white and red areas), whose values only change significantly towards the end of the simulation.

These sustained regular oscillations can be understood by comparing the energy capacity and growth rate of cells in different states (bottom panel). When one gene is fully inactivated, the energy capacity in the cell falls below that in cells with no mutations (black dashed line). As a result, there is less energy capacity to sustain protein expression, which causes the growth rate to drop compared to cells with no mutations (green dashed lined). Cells with one gene inactivated will therefore get removed from the turbidostat at a faster rate than E-cells. In other words, while transitions from the E-state to directly connected states are possible, there is little selective pressure to do so, explaining why stable oscillations persist over an extended period of time. As M-cells have the highest growth rate in the population, they gradually become dominant relative to other cell types. As a result, a tipping point is eventually reached whereby M-cells out-compete other cell states.

The reported drop in growth rate is initially counter-intuitive, as one may expect cells to gain more resources for growth-supporting processes when inactivating any one genes. A plausible explanation is that, for cells with no mutations, sustaining a cycle of mutually repressing species prevents any one protein from reaching high expression levels and thus prevents an excessive drain on cellular resources. By removing one component and thereby breaking the repression cycle, one protein is able to become dominant and, in turn, causes a higher drain on resources compared with when all three genes are active.

Mutation dynamics can be analysed in greater detail by additionally considering the effects of partially-inactivating mutations on each gene's promoter. The top panel in Fig. 4c shows how this can be achieved using the $[s_3, d_3]$ framework, with construct designs associated with select states being shown. In contrast to Fig. 4b, the introduction of partially-inactivating mutations causes the regularity of oscillations to change from the offset, alongside an immediate reduction in the quantity of E-cells. This behaviour can again be understood by comparing the energy capacity and growth rate between different states. The bottom panel shows that, when mutation events partially inactivate the promoter of one gene, the cell's energy capacity oscillates at a higher average value than with no mutations. In turn, the greater availability of resources allows for the expression of proteins at higher levels, which boosts the growth rate of cells with partially-inactivating mutations cells relative to E-cells. This suggests that reducing the repression of any protein, while not totally inactivating it, is selectively advantageous and allows the corresponding cell types to accumulate in the turbidostat at a faster rate. The erratic nature of the oscillations observed later in the simulation can be explained by comparing the protein dynamics in different states, as is outlined in Supplementary Note Section S3.2.

The higher energy capacity resulting from partially-mutated genes can again be explained by considering the strength of repression within different gene constructs. Partially reducing the expression of one gene reduces its drain on the cell's resources. However, its remaining activity prevents the other protein species from dominating to excessive levels. The resulting sum of repression is, therefore, sufficiently low such that the cell's available energy capacity increases. Contrasting this with the effects of fully-inactivating mutations, our results suggest that synthetic repressilators are typically resistant to single points of failure.

## Discussion

Our modelling framework has been designed to model evolution in synthetic constructs with variation in genetic parts, number of genes, and feedback regulation. Once a user specifies their system's design and the degree of mutation granularity to explore, equations are generated that govern transitions between different mutation phenotypes, taking into account (i) sequence-dependent mutation probabilities and (ii) gene expression-dependent growth rates. In order to model how each mutation phenotype is connected to one another, we position states associated with a different mutation location along a new orthogonal axis, and states associated with a different mutation severity along equivalent axes.

In outlining our framework, we use different mutation locations to resemble distinct functional parts, such as the promoter or the RBS of a gene construct. In doing so, a natural question arises about the granularity of functional parts: with a promoter, for example, subtly different functional effects could result from mutating its '−35', '−10' and spacer regions, thereby necessitating different mutation states for each. Taking this to the extreme, one could in theory assign a mutation state to each nucleotide in a synthetic construct and explore the resulting evolutionary dynamics through an extremely high-dimensional space. Scaling frameworks to this level of resolution would create more problems, however, such as the impossibility of predicting the effect of mutating each nucleotide, or the challenge of determining how insertions and deletions would be represented. These themes are the subject of discussion by[6] who theorise how predictive evolutionary landscapes can be designed, suggesting that the sequence space of different mutation types can be navigated by forming probability distributions for a device's fitness and its function. At the other end of modelling granularity, a synthetic construct could instead be considered as either wholly active or wholly inactive, rather than dissecting it into individual genetic parts. In many scenarios, this effect can be approximated by mutating a gene's promoter region, as doing so would block all downstream gene expression. This effect is utilised in our analysis, where we approximate the complete inactivation of gene constructs by modelling mutations to their promoters.

Regardless of how distinct functional parts are defined, exploring mutations with a part-driven approach strikes a balance between model complexity and usability. Despite this, one may want to consider the computational cost associated with our framework when varying the number of mutation locations and severities. As proxies for model complexity, we can consider the total number of states ('$n$') and the total number of state transitions ('$T$') in a system, which can be calculated as '$n = s^d$' and '$T = \binom{s}{2} \cdot d \cdot s^{d-1}$', respectively (Section S5). Given that each mutation state uses an entire set of host-aware cell equations (Section S1.1), it is recommended that care is taken in first determining the most relevant mutagenic parts of a system in order to minimise computational cost and to simplify the model output. Further detail can be added after this, if desired. For example, modelling each mutable part with only one intermediate state might be sufficient to capture the mutation heterogeneity present in a system, or modelling mutations to just the promoter and not the RBS might produce the desired dynamics. The capabilities of modelling at these levels of granularity are shown in our results, where we uncover unique design considerations of genetic constructs using frameworks that consider only promoter mutations with one intermediate mutation state.

Additional variation in mutation dynamics can also be added by modelling gene expression in more detail. The current cell equations, based on the model by[29], describe transcription and translation elongation as single-step processes, and in doing so allow users to change the values of parameters associated with a gene's maximum transcription rate and mRNA-ribosome binding rate. Despite this, the equations do not describe other core components like the coding sequence (CDS), terminator, and origin of replication, and therefore cannot capture the effects of codon efficiency, termination efficiency, and copy number. Adding variation to the CDS's function, for example, could be important when considering variable codon design. This is because inefficient codons are known to cause ribosomal traffic jams on mRNA transcripts[51–53], thereby hindering the translation of other proteins that are essential for cell survival. Some existing models of translation consider these effects in detail[34,53–55], with[34] additionally modelling their impact on shared cellular resources and cellular growth rate. In addition to modelling genetic parts in greater details, variation could be added by specifying different energy sources, adding other proteome fractions for the effects of transcription factors, or including more resource pools for other metabolites like tRNAs and NTPs. These may be particularly important when capturing other consequences of high protein expression, such as the accumulation of toxic metabolites[56] or misfolded proteins[57,58]. Some host-aware models capture these effects well[31,32], and so just like with modelling the CDS in more detail, they could in theory be accounted for in our mutation-aware framework.

In our results we showed how our model can be applied to a number of use cases that could aid researchers across many areas of cell engineering. First, we showed how it can be used to model protein yield and genetic shelf life. This is inherently useful when trying to optimise systems for function in the biotechnology industry, where reagent costs and experimental reliability are often of large concern[4]. Our analysis during the section on protein production dynamics showed how the design of synthetic constructs impacts both yield and shelf life via the maximum transcription rate '$\alpha_E$' and the mutation probability '$z_M$', both of which can be altered by changing a construct's promoter or its nucleotide sequence[22], respectively. When optimising yield, we showed how the time scale of experiments impacts the choice of '$\alpha_E$' due to a trade-off between production capacity and selection pressure (Fig. 2b), a trend that is supported by recent experimental studies[26]. When improving shelf life, we see that for many parameter values, changing '$\alpha_E$' becomes increasingly more impactful than changing '$z_M$' when more mutations accumulate (Fig. 2c). Growing populations to high degrees of mutation is not typically viable; however, these results may empower researchers to justify one experimental change to their synthetic construct over another. This becomes more important when either time or resources are limiting, as it could be unfeasible to modify both a construct's parts (to change its growth rate, for example via '$\alpha_E$') and its sequence (to change its mutation probability).

To better illustrate the potential of our framework, we then explored the utility of varying the granularity of mutation dynamics of the genetic toggle switch and the repressilator. Both of these motifs have extensive theoretical underpinnings; however, their mutation-driven behaviour has been far less explored. First, for a genetic toggle switch, we explored the impact of mutations by monitoring how its key trait is affected: its ability to switch from one stable state to another. Intuitively, it was found that the onset of mutations decreased the construct's switching capacity; however, more interesting observations were made when exploring the effect of partially inactivating mutations. When permitting intermediate mutations, we found that its capacity to switch between stable states can temporarily increase (Fig. 3). In these instances, less inhibitor is required to induce switching which could have varied experimental implications. Inducing switching behaviour with less reagent could deliver a simple economic benefit; however, a higher switching capacity may make toggle switches more prone to unintended switching, for example due to leaky gene expression. While these ideas have not been experimentally tested, their implications could be important for applications that require precise and predictable switching behaviour. As shown in[59], for example, systems engineered to decouple cell growth from synthetic product formation need to switch gene expression precisely in order to manage cell toxicity and to not waste product formation.

Following this, we applied our framework to the mutation-driven-dynamics of a repressilator. As expected, we found that the oscillations of protein species decayed over time; however, our simulations also suggest that repressilators are resistant to single points of failure, such that such that completely inactivating any one gene is selectively disadvantageous (Fig. 4). We explain this by comparing the cell's energy capacity when in different mutation states, finding that it drops when removing individual genes and thereby leads to a drop in growth rate. While the mechanistic details of this have not been experimentally tested, its foundational idea is supported by data from[26] who show that increased synthetic gene expression is correlated with higher gene expression burden, lower cellular growth rate, and a higher selection pressure for mutant cell phenotypes. Data that focus specifically on repressilators are more scarce; however, some interesting comparisons can be made from recent studies.[60], for example, experimentally show that decreasing sources of noise in their repressilator designs improved the regularity of oscillations, while[61] simulations suggest that positive autoregulation plays an important role in maintaining oscillation robustness. As a result, while our analysis suggests one source of irregularity, such as mutation heterogeneity, other factors may be important to consider when producing oscillation-driven applications.

The applications above show how new paradigms for synthetic construct design could be used to enhance many synthetic devices. The modularity of our framework means that these analyses can be extended to any gene regulatory network and could potentially uncover other insights into their evolutionary dynamics. For example, the mutation dynamics of feedforward loops[15] or those of motifs embedded in larger networks[62] would be unobvious a priori, and would therefore constitute interesting candidates to analyse using our framework.

## Methods

### Numerical implementation

The modelling framework is constructed as outlined in the main text and is fully detailed in the supplementary information (Supplementary Note Section S1). All code was written using MATLAB R2020a. ODEs were solved using the stiff numerical solver ode15s with a relative tolerance of $10^{-6}$ and an absolute tolerance of $10^{-9}$. For cases where mutations were permitted, simulations were run starting with all cells in the E-state and continued until all cells reached the M-state. When simulating each framework, a prior simulation was run to obtain the initial values of variables for each state. At the end of each prior simulation, the last values of each variable were used as the initial values for the variables in the main simulation. For the models exploring protein yield and cell viability (Fig. 2), the prior simulation was run without any synthetic construct expression by setting values of $m_H$, $c_H$ and $H$ to zero. For the models exploring the toggle switch dynamics (Fig. 3), the prior simulation was run by setting values of $m_H$, $c_H$ and $H$ to an arbitrary positive value for protein-A and to zero for protein-B. In addition, the prior simulation was run without an inhibitor, as we wanted the main simulations to begin with a fixed concentration of protein-A. For the models exploring the repressilator dynamics (Fig. 4), the prior simulation was run with arbitrary values of $m_H$, $c_H$ and $H$ for each synthetic gene, with different values for each gene to ensure that oscillations were produced. Full details of how the model is constructed are given in Supplementary Note Section S1.

### Analysis of model fitting

We fit our model to data from ref. 17. The values of data points were inferred using the online tool 'WebPlotDigitizer' (version 4.6). Simulation results were fit to each data set by systematically varying the core mutation parameters ($\alpha_E$, $z_M$, $\alpha_I$, $z_I$) for both $[s_2, d_1]$ and $[s_3, d_1]$ frameworks until the closest fit was obtained, evaluated by the smallest root mean square deviation. Values for all fits are given in Supplementary Note Section S2. As part of the analysis in this section, we also compared our

parameters with 'relative instability predictor' (RIP) scores generated by the online tool 'EFM calculator' (version 1.0.1). The sequences used as input for the EFM calculator were derived by searching for each part that Sleight et al. report in their study within iGEM's 'Registry of Standard Biological Parts' and joining them together in the correct order and orientation. The sequences are provided in Supplementary Note Section S2.2.

## Reporting summary

Further information on research design is available in the Nature Portfolio Reporting Summary linked to this article.

## Data availability

The data used in Fig. 1d and Fig. S2 are from the study of ref. 17 and can be found at: https://doi.org/10.1186/1754-1611-4-12.

## Code availability

All code used to implement the models can be found at the following Github repository: https://github.com/ddmingram/Mutation-aware-model.

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

## Acknowledgements

D.I. acknowledges support from his Wellcome Trust PhD studentship (grant ref. 203953/Z/16/A). G.-B.S. acknowledges support from his UK EPSRC Fellowship for Growth in Synthetic Biology (grant ref. EP/M002187/1) and his UK Royal Academy of Engineering Chair in Emerging Technologies for Engineering Biology (RAE CiET1819\5).

## Author contributions

D.I. co-designed the project, wrote the model, conducted the analyses, co-wrote the manuscript, and created the figures. G.-B.S. co-designed the project, supervised the project, and co-wrote the manuscript.

## Competing interests

The authors declare no competing interests.
