## [Peer Review File · Nature Communications]

Reviewers' Comments:

Reviewer #1:

Remarks to the Author:

The authors describe a flexible approach to predict how mutations will “arise and propagate” in a population of engineered cells, and the dynamics with which these mutations may affect the engineered synthetic circuits. The authors then model two specific circuits (the toggle switch and repressilator), simulate these models, and suggest qualitative hypotheses about how resilient populations of these circuits are to mutations, and how the behavior of these circuits will change as mutations accumulate.

While the problem addressed is a critical one, and the approach taken seems reasonable, the main weak point of this work is the lack of experimental evidence to validate their framework. The authors do some fitting of their model to experimental data, but the retrodictive nature of the analysis along with the large potential for overfitting make this analysis an insufficient replacement for experimental prediction.

Novelty: the approach is innovative in that to this reviewer’s knowledge (including a brief literature review) there is no well-established, general, computational framework for predicting how mutations will arise and spread in a population of engineered cells that takes into account the burdens of the circuit upon cell growth rate, and how this will ultimately affect the circuit behavior. The authors description of related works on prediction of how mutations will be introduction to and spread throughout populations, in the introduction, however, is extremely short (basically only two citations). It would be worth spending more time discussing prior literature on the subject. For example, this more recent reference may be one to further investigate:

Nikolados, Evangelos-Marios, et al. "Growth defects and loss-of-function in synthetic gene circuits." *ACS synthetic biology* 8.6 (2019): 1231-1240.

There is also a lot of work in population systems biology on stochastic process models for mutants (e.g. Moran model, clonal interference). It could be good to mention some of this work and why those models are insufficiently simple for the problems synthetic biologists may wish to solve, as to better connect the authors’ current efforts with past ones in the field.

Significance: if the authors can demonstrate that this modeling framework is highly predictive in experimental settings, it would have a significant effect on the field of synthetic biology. The main issue with this paper, however, is that demonstrating this predictiveness really requires testing against experimental data (not just fitting a model retrospectively to data, especially given possible overfitting). Without experimental data, the significance of this work is highly reduced. Additionally, it is not clear how to choose the modeling parameters without careful experimental study. I am concerned that even if the model would be predictive when the “true” parameters of the system are known, finding these parameters in practice could be a large problem. In practice, would fitting the models to experimental data be enough for finding good parameters without overfitting? This issue is left unaddressed, leaving the work potentially significant in theory, but possibly lacking direct impact in its current form.

Analysis: One place upon which the simulation-based analyses/predictions could be improved: In the SI, many parameters lack sources. Are the chosen parameters realistic? Do the conclusions from simulations for the toggle switch and repressilator change if they are modified? These questions ought to be addressed in the SI.

Presentation: The authors have very nicely made figures. For clear communication, I think it could be helpful to spend slightly longer in the main text explaining what the X and ψ variables from equation (5) represent. Thanks to the SI, this is not a big problem, but it reduces readability to need to reference the SI for clarity in the middle of the method description. Also, there is a small typo (an unwanted space) in the title of section 4.2.3.

Reviewer #2:

Remarks to the Author:

The manuscript attempts to answer an important question of the effect of mutations, the resource competition between synthetic circuit and host on the dynamics of the synthetic circuit through a phenomenological model. The authors have done extensive analysis of how a mutation (partial or complete) rate and the expression rate of the synthetic constructs are able to affect the dynamics of the circuits over time, explaining the possible reasons for the loss of functionality observed in these constructs over time. Overall, the paper is well written, easy to follow and tries to address an important question about why synthetic circuits cannot maintain their intended dynamics in cultures over long periods of time.

Here are some suggestions for the authors to further improve their manuscript:

1. Could the authors briefly explain about Weisse et al.'s framework, specifically the host-aware cell equations, in the main text itself, for better readability?
2. Currently, the authors have considered mutations leading to partial and complete inactivations. Could the current framework also take into account mutations which can lead to increased transcription rate of a gene? How would it affect the dynamics of the example motifs (toggle switch and repressilator)? Since the model is incorporating the stress caused by overexpression through "host-aware" equations, would these cause a detrimental effect to their dynamics, if so how?
3. In the results section on repressilator, when the feedback loop (through inactivation of a gene) of the repressilator breaks, the overexpression of the dominant protein leads to reduced energy capacity and a subsequent decrease in growth rate. Could the authors provide more evidence to support this hypothesis? Would restricting the expression of the two active genes increase the growth rate of the mutated cells?
4. Can the authors comment discuss their results on the impact of mutations in the context of functional resilience of these network motifs considered (PMID: 36551270)?

Reviewer #3:

Remarks to the Author:

This manuscript presents a modelling framework to explore the consequences of mutations in synthetic genetic constructs on the productivity of the construct and the viability of cells carrying it. The authors specifically focus on mutations that impact expression levels of genes in the construct and examine the evolutionary fate of the mutations that reduce or completely remove the transcription of one or more genes in the synthetic construct. They use their model to investigate the evolutionary dynamics in three synthetic systems of increasing complexity, carrying one, two or three components. I think there is certainly utility for the presented modelling framework, but I have major concerns about its interest to a wider audience outside the synthetic biology community.

My major concern is that the authors do not account for any existing works that have very similar goals and objectives as the presented manuscript. Namely, the extensive literature from the fields of evolutionary biology and population genetics presented numerous models that have been developed to study how populations respond to mutations. For example, the formulation of the basic aspects of the model (presented in section 2) are very similar to Gillespie 1983 model of strong selection-weak mutation. Many concepts central to the presented model have been extensively modelled before in more detail than this manuscript could realistically achieve (simply because population genetics is a long-standing field and 1000s of papers on the topic have already been published). An example is that authors explore the effects of 'mutation heterogeneity' – a feature of most approaches to modelling evolution that are captured through the 'distribution of mutational effects'. In fact, very few models on the topic do not account for the fact that mutations can have different effects. Another example is the modelling of dynamics in a

turbidostat under restocking conditions – a scenario very similar to models of evolution under migration or source-sink regimes.

Due to this manuscript not being rooted in evolutionary literature, it seems to develop new terms for phenomena that have already been described. Some examples include: “mutation probability”, which is typically referred to as mutation rate; the description of mutational effects presented in Fig.2b, which is typically referred to as fitness landscape; “mutation spread”, which is closely related to the concept of fixation probability; “mutation severity”, which is typically referred to as fitness effect, etc. Most importantly, the authors do not mention the term ‘fitness’ once in the manuscript – a term that is central to the mathematical understanding of how populations respond to selection.

This is not to say there are no novel ideas presented in this paper. In particular, population genetics and evolutionary models do not typically concern themselves with mutations affecting gene expression levels, and there are some key and interesting differences that arise from considering regulation as opposed to protein function. For example, the distributions of mutational effects in promoters are different to those in proteins, and this can affect evolution. The authors could use some of the recently published models linking promoter sequence to expression levels (examples include : Kinney et al. PNAS 2010; LaFleur, Hossain and Salis bioarxiv 2021; Lagator et al. eLife 2022) to account for the biophysically realistic differences in the effects of mutations in promoters. Similarly, accounting for how promoter mutations affect RNA polymerase versus transcription factors that bind to the promoter could result in an interesting model that I have not encountered before. Having said that, I am surprised that the authors worked under the assumption that most mutations that remove the function of the synthetic construct affect regulation, as opposed to introducing a mutation that knocks out protein or transcription factor function (as the mutational space for such mutations is typically much larger). Exploring the differences between these types of mutations could be really interesting.

However, the general approach, which disregards existing literature from the field of evolutionary biology and population genetics, prevents this work from having broader appeal I think is needed for publication in a broad readership journal like Nature Communications. I would recommend targeting a journal more focused on synthetic biology, as the concerns I expressed above would be less relevant to that audience.

Dear Editor and Reviewers,

We thank the reviewers for their constructive comments, which we address point by point below. To help us thoroughly address all comments, we have split up the reviewers' comments into standalone numbered sections. Some responses relate to text changes, which we have highlighted in red in our revised manuscript. In particular, following the reviewers' suggestions we have extended our framework to broaden its scope, and completely rewritten the introduction and large parts of the discussion sections to better integrate our results and further clarify our contributions with respect to the existing literature. These changes have allowed us to significantly improve the quality of the manuscript beyond its initial submission, which we hope will be to the reviewers' satisfaction.

Reviewer #1

The authors describe a flexible approach to predict how mutations will "arise and propagate" in a population of engineered cells, and the dynamics with which these mutations may affect the engineered synthetic circuits. The authors then model two specific circuits (the toggle switch and repressilator), simulate these models, and suggest qualitative hypotheses about how resilient populations of these circuits are to mutations, and how the behaviour of these circuits will change as mutations accumulate.

While the problem addressed is a critical one, and the approach taken seems reasonable, the main weak point of this work is the lack of experimental evidence to validate their framework. The authors do some fitting of their model to experimental data, but the retrodictive nature of the analysis along with the large potential for overfitting make this analysis an insufficient replacement for experimental prediction.

Novelty: the approach is innovative in that to this reviewer's knowledge (including a brief literature review) there is no well-established, general, computational framework for predicting how mutations will arise and spread in a population of engineered cells that takes into account the burdens of the circuit upon cell growth rate, and how this will ultimately affect the circuit behaviour.

1. The authors' description of related works on prediction of how mutations will be introduced to and spread throughout populations, in the introduction, however, is extremely short (basically only two citations). It would be worth spending more time discussing prior literature on the subject. For example, this more recent reference may be one to further investigate: *Nikolados, Evangelos-Marios, et al. "Growth defects and loss-of-function in synthetic gene circuits." ACS synthetic biology 8.6 (2019): 1231-1240.* There is also a lot of work in population systems biology on stochastic

process models for mutants (e.g. Moran model, clonal interference). It could be good to mention some of this work and why those models are insufficiently simple for the problems synthetic biologists may wish to solve, as to better connect the authors' current efforts with past ones in the field.

Response

We thank the reviewer for noting our limited reference to prior work on mutation spread. When reviewing our introduction, we realised that more care could be taken to establish a firmer basis for how early work in population genetics has paved the way for our current understanding of evolution in synthetic biology. This was not as simple as just adding in another paragraph, and so we decided to completely reformulate the introduction to account for a more logical flow of information. This includes the following new order of information:

- Introducing why the topic of mutation spread in synthetic systems is of broad scope and interest;
- Describing how mutation spread has been historically studied, including reference to the models of Wright, Fisher, Moran and others;
- Explaining why, despite existing models, there remains a gap when addressing evolution in modern-day synthetic systems;
- Detailing the current state-of-the-art on predicting evolution in synthetic systems and explaining why it's limited in scope;
- Suggesting how existing work could be improved, leading to the methods covered in our paper.

As part of this new formulation, we also include a citation to Nikolados et al.'s work on the interplay between construct and host. For the optimal flow of information, and considering the other material we have added, we felt that this was best placed as a reference in our paragraph on host-aware models.

2. Significance: if the authors can demonstrate that this modelling framework is highly predictive in experimental settings, it would have a significant effect on the field of synthetic biology. The main issue with this paper, however, is that demonstrating this predictiveness really requires testing against experimental data (not just fitting a model retrospectively to data, especially given possible overfitting). Without experimental data, the significance of this work is highly reduced. Additionally, it is not clear how to choose the modelling parameters without careful experimental study. I am concerned that even if the model would be predictive when the "true" parameters of the system are known, finding these parameters in practice could be a large problem. In practice, would fitting the models to experimental data be enough for finding good parameters without overfitting? This issue is left unaddressed, leaving the work potentially significant in theory, but possibly lacking direct impact in its current form.

Response

This is an important point to address, so we thank the reviewer for highlighting it. The remit of our paper was to create a modular and expandable mathematical modelling framework that links the allocation of shared cellular resources to many areas of mutation and selection, including: measuring yield and cell viability, generating robust construct designs, and approximating the degree of mutation heterogeneity in a system. In doing so, we show how fitting to experimental data can be practically achieved, and offer an analysis about how the key mutation parameters can be interpreted. In this light, it is important to stress that our tool is applicable to various synthetic systems, giving the user flexibility to probe different degrees of mutation dynamics and augment other features they may wish to explore in the future. Our goal is *not* to rigorously identify parameters (perform system identification) but rather to provide the community with a much-needed framework to model evolution in populations of engineered cells. Experimental validation and system identification would require extensive experimental work including long-term culturing and characterisation of growth rates and genetic construct function in engineered cells, combined with sequencing of the engineered constructs over time. While this would be desirable for any study on evolution, it would involve highly extensive experimental set-up and sequencing.

Despite this, we would argue that our analysis does demonstrate *some* predictive capabilities. The four mutation parameters we focused on (α_E , z_M , a_i , z_i) each change the mutation dynamics in qualitatively different ways, suggesting that their combined effect can be disentangled from the resulting dynamics. To support this information, we discuss how each parameter can be experimentally interpreted below:

- α_E : maximum transcription rate. Higher values increase the maximum expression level of synthetic protein but compress the overall period of protein expression due to the penalty on growth rate. If a user has synthetic constructs with a range of promoters from the same library, then they can find a relationship between promoter strength and α_E by equating the maximum expression of one construct to the corresponding value of α_E .
- z_M : probability of a function-inactivating mutation. Higher values cause the protein production decay to begin at earlier time points. When interpreting this parameter, we compared our results to those of Jack et al.'s EFM calculator, a tool that links sequence composition to mutation rate. As part of their paper, Jack et al. analyse the mutation rate of the construct T9002-E and generate a value roughly equivalent to ours. We discuss this in Section S2 and explain why there are discrepancies when using the EFM calculator for the other T9002 variants. Accurately predicting z_M across a variety of sequences and experimental contexts will likely remain a significant challenge for years, and so to our knowledge, there is still no foolproof quantitative gauge for this parameter. One suggestion, therefore, is that users consider a combination of Jack et al.'s tool and our model before evaluating their specific use-case.

- **α_i and z_i** : maximum transcription rate of the intermediate mutation state (α_i), and mutation probability of creating an intermediate mutation state (z_i). In reality, a system will contain many different mutation pathways with subtly different effects. When using parameters for intermediate mutations, therefore, we are *not* suggesting that these should be interpreted as relating to one specific mutation pathway. Firstly this would be wholly inaccurate, and secondly, testing this experimentally would require an extreme level of sequencing detail. Instead, it is more fitting to interpret the combined effect of these parameters as the ‘degree of mutation heterogeneity’ in a system, which we loosely define as the extent to which mutation dynamics deviate from the foundational E-to-M mutation pathway. In turn, systems with lots of mutation variation will be best modelled by combinations of α_i and z_i that lead to varied dynamics through the presence of a sustained intermediate cell population, regardless of how that mutation variation is manifested. During our analyses, we show that the use of α_i and z_i allows a user to fit a wide variety of mutation dynamics. However if a user feels that their experimental dynamics are still not sufficiently captured, they can then explore the effects of modelling more than one I-population with additional intermediate mutation states and associated parameters for α_i and z_i . Given the interpretation of α_i and z_i , it may be advantageous in the future to provide some form of ‘mutation heterogeneity score’ which captures the extent to which dynamics deviate from the standard E-to-M pathway during the course of a simulation. This would likely be some combination of how the population’s cell growth and heterologous protein production differ from equivalent simulations using a two-state system.

With these details in mind, we feel that useful information can be derived from our simulation results, especially considering the scant alternatives in the literature at present. During our comparisons with experimental data, we re-evaluated the model complexity required for each new fit and found that, in some cases, modelling mutation heterogeneity (and hence using α_i and z_i) was *not* required to achieve close fits. We therefore feel that appropriate usage of our model strikes a balance between the ‘number of parameters’ and the ‘ability to predict complex behaviours’, or in other words between model complexity and prediction capability. In turn, we hope it is also evident why obtaining a *complete* predictive link is beyond the scope of this paper.

In formulating our reply to this point, we felt that prospective readers may have similar queries to the reviewer, and so we have added extra information about the use and interpretation of our parameters in the main script:

- In general we have been more careful with our usage of terms “fit”, “model” and “predict”, and avoid using the latter when it is not relevant. Most notably, this includes changing the word “predicting” in our paper’s title to “modelling”.
- In Section 2.2, after introducing how mutation severity is captured, we added a paragraph to highlight how our core mutation parameter should be interpreted.

Here, we give particular focus to how α_i and z_i capture the degree of mutation heterogeneity present in a system.

- We removed the parts of ‘parameter interpretation’ from our discussion, as it is now better captured in Section 2.
- At the end of Section 2.3, we added more details that better qualify why our model strikes a balance between model complexity and prediction capability.

3. Analysis: One place upon which the simulation-based analyses/predictions could be improved: In the SI, many parameters lack sources. Are the chosen parameters realistic? Do the conclusions from simulations for the toggle switch and repressilator change if they are modified? These questions ought to be addressed in the SI.

Response

As the reviewer notes, many of the parameters listed in Table S1 lack sources, as previously indicated by the ‘–’ symbol. In revising our manuscript, we have given this section much more attention by qualifying all missing parts with additional information and streamlining the presentation of our sources. Part of this process involved removing the potentially-confusing column, “changes from Weisse et al.”, and placing any relevant information in the legend. We also now provide explanations in the SI for the additional parameters required during the toggle switch and repressilator analyses. The values for these parameters were also moved to the first section, making the overall flow more logical.

We direct the reviewer to our changes made in the SI, however to provide a fuller account, we have detailed our explanations below for each ‘previously unsourced’ parameter from Table S1. These can be split into (i) those that relate to the design of the synthetic construct, (ii) those that relate to the growth conditions of experiments, and (iii) those that govern the host-aware reactions within the cell. We also note whether our simulation results would be affected by changes to these values where appropriate.

I. Parameters relating to the design of the synthetic construct

For some of these we explored the impact on mutations over a range of values, as we wanted to avoid restricting our analyses to individual systems and instead cover the effects of a broad range of synthetic construct designs.

- Maximum transcription rate in different states (α_x): we explored the effects of varying this over ranges that spanned many orders of magnitude. This matches the range of promoter strengths used in typical libraries, and we now provide an example source for this in Table S1.
- mRNA-ribosome binding rate for the H-fraction in different states (β_x): this was kept equivalent to the mRNA-binding rate of other proteome fractions so as to not place particular emphasis on H-translation.

- Length of synthetic protein (n_H): chosen to be equivalent to the average size of cytosolic proteins in *E. coli*, but can be modified to reflect synthetic proteins of different sizes.
- Probability of sustaining different mutations (z_x): like α_x , we explored the effects of varying this over many orders of magnitude to match the range of mutation rates seen across different synthetic constructs. Examples for the wide range of mutation rates in synthetic constructs can be seen in many sources, such as the data considered by Jack et al.'s EFM calculator.

II. Parameters relating to the growth conditions of experiments

- Nutrient quality (n_q): we maintained this at 1 as we weren't interested in varying different media concentrations or nutrient quality in our simulations, however some users may wish to.
- Turbidostat cell density threshold (N): we maintained this at 10^9 to resemble a dilute population of *E. coli* cells that is not space-limited. The value can be changed to reflect turbidostats with different cell thresholds - doing so would change the time scale over which mutation dynamics are observed, but would not affect the qualitative shape of the dynamics due to how the dilution function is implemented (Equation 1).

III. Parameters governing the general cell behaviour

The justifications for these values are generally taken from Weisse et al. The main implication of changing these values would be a change in the cell's growth rate, and as such the speed at which selection propagates in a system may differ, but the underlying patterns we observe in our analyses would not.

- Hill coefficient for the auto-inhibition of the mRNA within the 'housekeeping' proteome fraction (h_0): as the use of a regulated 'housekeeping' fraction is a simplified account of how *E. coli* regulates its cytosolic proteome, there is not an appropriate literature-based parameter value for this process. Any value that leads to sharp auto-inhibition is therefore appropriate, such as '4' in this case.
- mRNA-ribosome binding rate for R, Z, and Q fractions ($k_{R,Z,Q}^+$): a value of $60 \text{ cell h}^{-1} \text{ molec}^{-1}$ is used in Weisse's model in order to be "near the diffusion limit", i.e. the rate at which reactants move within the cell. For this to make sense, we added more detail in the last paragraph of Section S1.1, which now states that (i) all variables are measured in molecules per cell, and (ii) a cellular volume of $1 \mu\text{m}^3$ is assumed (approximately that of *E. coli*) for reactions that depend on the concentration of molecular species.
- mRNA-ribosome unbinding rate for R, Z, Q and H fractions ($k_{R,Z,Q,H}^-$): when modelling reversible reactions, it is standard to consider the unbinding rate to be equal to the binding rate, hence a value of 60 h^{-1} is used.
- Half-saturation constant for energy metabolism (K_e) and internal nutrient quantity (s): these values were chosen relative to each other in a ratio of 1:10 in order to

achieve varied energy metabolism dynamics when the quantity of enzymatic proteins (Z) varies.

4. Presentation: The authors have very nicely made figures. For clear communication, I think it could be helpful to spend slightly longer in the main text explaining what the X and ψ variables from equation (5) represent. Thanks to the SI, this is not a big problem, but it reduces readability to need to reference the SI for clarity in the middle of the method description. Also, there is a small typo (an unwanted space) in the title of section 4.2.3.

Response

We first outline the precise meaning of these variables at the end of Section 2.1 when introducing the set of equations for the two-state model (Equation (4)). Despite this, we agree that a reader would find it useful to have these definitions reinstated when consolidating all equations for the general model. We have added a sentence after Equation (5) to address this. Furthermore, we have improved the consistency of our notation throughout, particularly when using non-italics vs. italics to denote cell state labels vs. quantities of cells within states.

Reviewer #2

The manuscript attempts to answer an important question of the effect of mutations, the resource competition between synthetic circuit and host on the dynamics of the synthetic circuit through a phenomenological model. The authors have done extensive analysis of how a mutation (partial or complete) rate and the expression rate of the synthetic constructs are able to affect the dynamics of the circuits over time, explaining the possible reasons for the loss of functionality observed in these constructs over time. Overall, the paper is well written, easy to follow and tries to address an important question about why synthetic circuits cannot maintain their intended dynamics in cultures over long periods of time.

Here are some suggestions for the authors to further improve their manuscript:

1. Could the authors briefly explain about Weisse et al.'s framework, specifically the host-aware cell equations, in the main text itself, for better readability?

Response

Given the reliance of our framework on Weisse et al.'s host-aware model, we understand the desire to have access to its details during the main text. We are conscious that the main manuscript retains a strong focus on the new contributions that our work brings, i.e. how we integrate Weisse's model into our mutation-aware framework, however we agree that more detail could be integrated into the main text. In this light, we have added information in Section 2.2 after introducing Weisse's model that: (i) describes that the proteome is finite, (ii) lists the four fractions present in the proteome, and (iii) mentions that protein expression requires using energy from a finite supply. We also add another sentence at the end of the paragraph that directs the reader to the appropriate SI section for full details.

2. Currently, the authors have considered mutations leading to partial and complete inactivations. Could the current framework also take into account mutations which can lead to increased transcription rate of a gene? How would it affect the dynamics of the example motifs (toggle switch and repressilator)? Since the model is incorporating the stress caused by overexpression through "host-aware" equations, would these cause a detrimental effect to their dynamics, if so how?

Response

In our analyses so far, we have focused on mutations that decrease the activity of a part and have explored the burden-driven consequences of these. It is indeed conceivable, however, that mutations could increase a part's activity, such as is the goal of many 'directed evolution' experiments. Our framework can easily consider such mutations by simply specifying that a part has a higher activity than in the E-state when a partial mutation occurs. This is achieved by modifying the appropriate value within the input

vector ' $[p]$ ' (Figure S1). Using this idea, frameworks can now be represented by placing function-decreasing states downstream of the E-state and function-increasing states upstream of the E-state, with the latter type being distinguished using blue shading and the symbol ' M^* '.

Framework structures for mutations that both decrease and increase the activity of synthetic parts.

States that depict cells with a function-decreasing mutation are coloured in red and positioned downstream of the E-state, while those with a function-increasing mutation are coloured in blue and positioned upstream of the E-state. Frameworks for the first three 'dimensions' are shown. Some states in the d_3 framework are removed for clarity. Some areas between states are shaded to help visual clarity.

For the new framework structures shown in the figure above, the total number of states in each is the same as in our previously-depicted s_3 frameworks. Despite this, as it is now possible to transition from a state with lower activity to one with higher activity, it introduces the question as to whether reverse mutations should be modelled. For example, after transitioning from the E-state to the M-state, could we then transition from the M-state to the M^* -state, or even back to the E-state again? And if so, what would the probability of such transitions be? Answering such questions with any reliability would require a lot of extensive knowledge about a construct's sequence and the potential for that sequence to change in many different ways. Such approaches may require implementing dynamic 'evolutionary landscapes', such as those discussed by Castle et al. in their paper 'Towards an engineering theory of evolution'.

Regardless of how transitions are implemented, the effect of function-increasing mutations on the resulting dynamics will likely be negligible. This is due to the phenomenon that we explored with the repressilator in Section 5: any mutations that increase the burdensome effects of synthetic gene expression will decrease the cell's growth rate and in turn cause it to be outcompeted in a growing population. Any subpopulation containing 'more active' mutations will therefore fail to increase in size to a significant number and thus not have a significant effect on the population's dynamics.

To show this, we plot example dynamics for the toggle switch and the repressilator. In each case, we use the same parameters as for our s_3 simulations, except now denoting the ‘intermediate’ parameters with the subscript ‘M*’ and setting $a_E < a_{M^*}$.

Modelling synthetic constructs that can sustain function-increasing mutations. (a) Toggle switch simulation. *Top:* as in Figure 3 from the main text. *Bottom:* relative growth rates are plotted for cells in three states: the E-state, the state where protein-A has increased activity, and the state where both protein-A and protein-B have increased activity. For each synthetic gene: $[a_E, z_M, a_{M^*}, z_{M^*}] = [10^5 \text{ molec h}^{-1} \text{ cell}^{-1}, 10^{-6}, 10^6 \text{ molec h}^{-1} \text{ cell}^{-1}, 5 \times 10^{-2}]$. **(b) Repressilator simulation.** *Top:* as in Figure 4 from the main text. *Bottom:* relative growth rates are plotted for cells in four states: the E-state, and states with one, two or three of the synthetic proteins having higher activity. For each synthetic gene: $[a_E, z_M, a_{M^*}, z_{M^*}] = [10^5 \text{ molec h}^{-1} \text{ cell}^{-1}, 10^{-6}, 10^6 \text{ molec h}^{-1} \text{ cell}^{-1}, 1.2 \times 10^{-2}]$. In both subplots, ‘*’ denotes a protein associated with a function-increasing mutation.

To show the dynamics for the synthetic constructs, we plot the average concentration of each synthetic protein per cell, and the growth rate relative to the M-state of various ‘states of interest’: the E-state, and states with different numbers of synthetic genes having increased activity as denoted by ‘*’. For both constructs, the protein dynamics remain relatively stable until the the point at which the M-state starts to dominate, similar to the dynamics in Figure 3b and Figure 4b when only severe mutations are modelled. This can be again explained by the growth rates of the states (lower panels) where there is an increase in protein activity: increasing the activity of any combination of the synthetic genes produces no gain in growth rate, and as such no selective advantage results.

Aside from considering how mutations change the activity of a synthetic part, we note that the effect of mutations may be linked to the function of the gene itself. For

purposes of generality, we have assumed that synthetic genes have no functional effect on the cell other than their drain on shared cellular resources. In other words, we have considered the effect on ‘fitness’ but not ‘utility’. If genes were instead modelled to additionally affect core cellular processes, then the consequences of synthetic gene expression would be less clear-cut. For example, if a gene increased the rate of energy metabolism, then increasing its activity would not only drain shared cellular resources, but also increase the activity of the cell. This could easily be implemented in our model by adding a positive autoregulation term that links the concentration of synthetic protein, H , to the rate of energy metabolism, $\varepsilon(Z)$, such that the modified rate of energy metabolism, $\varepsilon^*(Z)$, would be of the form:

$$\varepsilon^*(Z) = \varepsilon(Z) \cdot \frac{\left(\frac{H}{K_H}\right)^{h_H}}{1 + \left(\frac{H}{K_H}\right)^{h_H}}$$

where K_H and h_H denote the half-saturation constant and the hill coefficient, respectively. Similar effects could be applied to any of the cellular processes that our model considers, either as negative or positive regulatory effects. In turn, both fitness and utility could be modelled for any of the gene constructs we consider.

Given the extensive response that reviewer comment has generated, we felt that it would be beneficial to readers to include it as a standalone section in the SI. As such, we have placed the above content in a new section entitled ‘Modelling other types of mutation’ (Section S4). We also now make reference to this section in the main text, during the penultimate paragraph of Section 2.1. Finally, we now clarify that the inequality “ $\alpha_E > \alpha_i$ ” is not a requirement during Section 2.2.

3. In the results section on repressilator, when the feedback loop (through inactivation of a gene) of the repressilator breaks, the overexpression of the dominant protein leads to reduced energy capacity and a subsequent decrease in growth rate. Could the authors provide more evidence to support this hypothesis? Would restricting the expression of the two active genes increase the growth rate of the mutated cells?

Response

The mechanistic basis of the effect that the reviewer highlights is similar to effects explored in our Nature Methods paper in 2015 (‘Quantifying cellular capacity identifies gene expression designs with reduced burden’), which includes experimental results showing how increased heterologous gene expression leads to higher gene expression burden and lower cellular growth rate. This is a good basis to suggest that a cell’s burden-dependent energy capacity is important in deciding how its selection pressures evolve, and in revising our paper we now make reference to this in our discussion. As far as we know, however, there is no experimental evidence that considers the resource-dependent consequence of inactivating genes in a repressilator. Obtaining this would be very challenging, as it would require forcing a repressilator to mutate in a specific way while tracking the energetic cost and sequencing the resulting cell

phenotypes. It seems intuitive, though, that nature would be more likely to find more gradual routes of inactivating systems rather than removing whole components outright.

To answer the reviewer's second question, completely inactivating one gene and reducing the activity of the two remaining genes would indeed lead to a higher growth rate compared to cells with just one gene inactivated. This can be seen by overlaying the growth rate of cells within each state onto the repressilator's $[s_3, d_3]$ framework. We do this in the figure below, highlighting the states in question whose cells have relative growth rates of 0.37. This is in comparison to cells with just one inactivated gene (0.09) and cells in the E-state (0.13). In order to convey more information to the reader, we have now added more labels for 'relative growth rate' to the states in Figure 4c.

The $[s_3, d_3]$ framework for the repressilator. Relative growth rates for cells in all states are added. The states with one gene completely inactivated and the remaining genes partially inactivated are highlighted.

4. Can the authors discuss their results on the impact of mutations in the context of functional resilience of these network motifs considered: *Harlapur, P., Duddu, A.S., Hari, K., Kulkarni, P. and Jolly, M.K., 2022. Functional resilience of mutually repressing motifs embedded in larger networks. Biomolecules, 12(12), p.1842.*?

Response

The new publication that the reviewer highlights discusses themes which are highly relevant to our model, and so we have augmented our discussion to include a commentary on this. The study in question theorises how the dynamic behaviour of toggle switches and repressilators ("toggle triads") changes when embedded in larger networks of varying sizes and connectivity. We felt it was most relevant to discuss this after we suggest how our framework could be applied to many construct designs in the future. This is now present at the end of Section 6.2, following a reformatting of our whole discussion in response to another reviewer's comments. Accounting for larger networks would broaden the scope of our model, and in doing so we postulate how mutations would buffer the additional effects of these larger networks with a focus on the potentially-unintuitive effects of selection pressure. For example, the effects of larger networks may buffer the fitness cost associated with completely inactivating one gene within a repressilator.

Reviewer #3

This manuscript presents a modelling framework to explore the consequences of mutations in synthetic genetic constructs on the productivity of the construct and the viability of cells carrying it. The authors specifically focus on mutations that impact expression levels of genes in the construct and examine the evolutionary fate of the mutations that reduce or completely remove the transcription of one or more genes in the synthetic construct. They use their model to investigate the evolutionary dynamics in three synthetic systems of increasing complexity, carrying one, two or three components. I think there is certainly utility for the presented modelling framework, but I have major concerns about its interest to a wider audience outside the synthetic biology community.

1. My major concern is that the authors do not account for any existing works that have very similar goals and objectives as the presented manuscript. Namely, the extensive literature from the fields of evolutionary biology and population genetics presented numerous models that have been developed to study how populations respond to mutations. For example, the formulation of the basic aspects of the model (presented in section 2) are very similar to Gillespie 1983 model of strong selection-weak mutation.

Response

We thank the reviewer for highlighting the issues in how we contextualise our work. When reviewing our introduction, we realised that more care could be taken to establish a firmer basis for how early work in population genetics has paved the way for our current understanding of evolution in synthetic biology. This was not as simple as just adding in another paragraph, and so we decided to completely reformulate the introduction to account for a more logical flow of information. This includes proposing a stronger basis for why this work has wider relevance, and an account of how Wright, Fisher, Moran and others pioneered our understanding of 'mutation spread'.

In regard to the reviewer's second point, we note that the formulation of the equations in our two-state model is similar to those in historical models, including those analysed by Gillespie. However, we do not make claims to originality in this regard. Our points of contribution come from combining several previously disconnected aspects of predicting evolution in the context of synthetic biology: (i) resource-aware modelling, (ii) mutation heterogeneity, and (iii) flexibly accounting for varied synthetic construct designs. While these aspects are not by themselves novel, combining their features to model mutation spread is, and given the current state-of-the-art, we feel that it greatly enhances our ability to explore the evolutionary dynamics of synthetic systems embedded in living cells. We hope that our newly formatted introduction, particularly the ideas presented in the second half, make this clearer.

2. Many concepts central to the presented model have been extensively modelled before in more detail than this manuscript could realistically achieve (simply because population genetics is a long-standing field and 1000s of papers on the topic have already been published). An example is that authors explore the effects of ‘mutation heterogeneity’ – a feature of most approaches to modelling evolution that are captured through the ‘distribution of mutational effects’. In fact, very few models on the topic do not account for the fact that mutations can have different effects.

Response

We appreciate that population genetics has accounted for strong progress in modelling mutation spread in various contexts, and as outlined in response to the first point, we hope that our reformulated introduction now better accounts for this. We do not claim that specific modelling concepts such as ‘mutation heterogeneity’ are by themselves novel. Instead, it is the combination of modelling resource-aware cell dynamics, mutation heterogeneity, and modular synthetic construct designs that facilitates our core contribution. To our knowledge, no other existing model achieves this. We agree, however, that stronger parallels could be drawn between the concepts that we use in our work and how they are classically portrayed in population genetics. As a result, we have now added comparisons to terms from population genetics when introducing many of our core concepts. Some of these are discussed in response to ‘point 4’ below, but to address the example given in this point, we now include reference to how mutation heterogeneity is equivalent to the ‘distribution of effects of mutations’ during the introduction.

3. Another example is the modelling of dynamics in a turbidostat under restocking conditions – a scenario very similar to models of evolution under migration or source-sink regimes.

Response

We use a turbidostat as our modelling context because (i) this is routine when conducting many continuous culture experiments in synthetic biology, and (ii) it simplifies our equations by providing the assumption of constant cell density. We are careful to not make any claims that modelling in turbidostats, or indeed any forms of continuous culture, is novel. Nevertheless, we describe the equations for the turbidostat’s implementation in Section 2.1 as we feel this helps the reader understand how we piece together the fuller multi-state model in Equation (5).

4. Due to this manuscript not being rooted in evolutionary literature, it seems to develop new terms for phenomena that have already been described. Some examples include: “mutation probability”, which is typically referred to as mutation rate; the description of mutational effects presented in Fig.2b, which is typically referred to as fitness landscape; “mutation spread”, which is closely related to the concept of fixation probability; “mutation severity”, which is typically referred to as fitness effect, etc. Most importantly, the authors do not mention the term ‘fitness’ once in the

manuscript – a term that is central to the mathematical understanding of how populations respond to selection.

Response

We thank the reviewer for pointing out that we indeed forego many opportunities of using core terms from evolutionary biology. We have since revised the manuscript to carefully include links to some of these terms where relevant, so that a wider variety of readers can better understand how our mathematical formulation is rooted. In some instances, however, we feel that there is not a true 1:1 relationship between the terms used in our paper and the terms that the reviewer suggests, and explain this below.

- Mutation rate: the way in which we formulate our state transition equations requires bounding the values of the parameter z_X between 0 and 1 inclusive, making it a probability and not a rate. It is true that much experimental literature reports mutation events as ‘rates’, and so this needs to be addressed when performing model fitting using values from the literature. We encounter this problem when fitting our model to Sleight et al.’s experimental data. During Section S2, we provide an analysis on how our values of z_M can be compared to the ‘mutation rate’ inferred from tools such as the EFM calculator, and in doing so provide the following justification:

“Given our framework requires values as ‘probabilities’, we can calculate these from ‘rate’ values using the Poisson distribution: this gives a probability (p) of X events happening if they occur at some time-averaged rate, Λ : $p(X = k) = \frac{\Lambda^k \cdot e^{-\Lambda}}{k!}$. For the probability of a mutation occurring in one generation, $p(X = 1) = \frac{\Lambda^1 \cdot e^{-\Lambda}}{1!} = \Lambda \cdot e^{-\Lambda}$. When values of Λ are small, $\Lambda \cdot e^{-\Lambda} \approx \Lambda$, meaning we can use the following approximation ‘probability \approx rate’ as a reasonable assumption.”

As such, we intentionally use the term ‘probability’ over ‘rate’, and furthermore argue that our usage of “probability” is valid when interpreting literature values.

- Fitness: this is a crucial term in any study on population-scale mutation dynamics, and as such we have revised our manuscript to include it where necessary. Specifically, the ‘fitness’ of a particular mutation phenotype can be equated to our usage of ‘growth rate’, which is arguably the most important parameter in our model. Cells with higher growth rate will out-compete those with lower growth rate, and in turn are deemed to have a higher ‘fitness’. In order to embed this concept more to readers, we have added a sentence in Section 2.1 that explains the equivalence between growth rate and fitness.
- Fitness effect: having established a link between fitness and growth rate, we now feel that it is clear how our usage of ‘mutation severity’ conveys the idea of ‘fitness effect’. Here, mutations reduce the activity of a synthetic construct, and

via the redistribution of shared cellular resources, increase the growth rate of the cell. The greater the inactivation caused by the mutation (what we term as 'severity'), the greater the impact on growth rate. To avoid any doubt, however, we expanded a sentence at the start of Section 2.2 to explicitly link 'severity' to 'fitness effect'.

- Fitness landscape: this is an important visual tool when studying many aspects of mutation spread in engineered cell populations, however using our definition of fitness above (as equivalent to growth rate), it is not what we plot in Figure 2b. A fitness landscape would plot the growth rate of cells across the different parameter combinations of α_x and z_M , whereas we plot protein yield. We explore the relationship between growth rate and protein yield in Section 3, noting that cells producing synthetic protein at a slower rate will grow faster and sustain protein production for longer before being outcompeted by mutant phenotypes. This is, in turn, why the optimal region for protein yield suggests an intermediate value of α_E , rather than a minimum or maximum value. If we were to plot a true fitness landscape using growth rates, the brightest spots would indeed relate to the lowest α_E value.
- Fixation probability: i.e. the probability that a particular mutation phenotype will eventually reach unity in a population. This is indeed a more rigorous term than what we loosely describe as 'mutation spread'. Despite this, we use 'mutation spread' intentionally because the scope of our analysis covers a wide range of concepts such as: device failure, optimising yield over time, and uncovering robust synthetic construct designs. We therefore feel that if we referred specifically to 'fixation probability', we would mislead the reader in what we are trying to achieve, as we never attempt to calculate this.

5. This is not to say there are no novel ideas presented in this paper. In particular, population genetics and evolutionary models do not typically concern themselves with mutations affecting gene expression levels, and there are some key and interesting differences that arise from considering regulation as opposed to protein function. For example, the distributions of mutational effects in promoters are different to those in proteins, and this can affect evolution. The authors could use some of the recently published models linking promoter sequence to expression levels (examples include : Kinney et al. PNAS 2010; LaFleur, Hossain and Salis bioarxiv 2021; Lagator et al. eLife 2022) to account for the biophysically realistic differences in the effects of mutations in promoters. Similarly, accounting for how promoter mutations affect RNA polymerase versus transcription factors that bind to the promoter could result in an interesting model that I have not encountered before.

Response

The reviewer makes an interesting point about fine-graining our model to account for (i) different promoter designs, and (ii) transcription factors. Both of these aspects relate to

the granularity with which a system is modelled. In building our framework, it was vital to strike a balance between model complexity and prediction capability throughout, from the choice of our host-aware cell equations, to how variation in mutation dynamics is captured. To strike a balance, we choose to base the host-aware cell equations on Weisse et al.'s model, and model variation in mutation location through distinct functional units, such as an entire promoter or an entire RBS.

First, for the host-aware cell equations, more detailed cell models could have been used that consider gene expression in more detail, including the use of transcription factors and a focus on additional part components, such as the CDS. However, we wanted to use the simplest account of gene expression that allows probing the essential elements of gene construct design. These ideas were previously discussed in the discussion. Next, concerning the resolution of our mutation modelling, we note that there is ambiguity about how functional units are defined. For example, one could in theory dissect a promoter region into further subregions, such as its '-35', '-10' and 'spacer' regions. At the extreme, one could in theory use different locations for each nucleotide within a promoter, however doing so with our existing framework design would introduce numerous problems associated with model complexity. We briefly touched upon these ideas in Section 2.

Given the above, we therefore feel that our framework strikes a good balance. If users want to model additional processes, they could modify the host-aware cell equations to reflect this. In light of the reviewer's comments, however, we felt that a lot more attention could be given to these concepts, both in the discussion and with broader examples. As a result, we have made the following significant changes to the manuscript:

- We have significantly streamlined our discussion with the first subsection now being focused on how our model strikes a balance between model complexity and usability. We include more refined detail on the modelling granularity of both the host-aware modelling and the mutation modelling.
- We removed the last discussion subsection as we felt it was redundant, and its essential points are now better captured in our first discussion subsection.
- We added a new section in the SI (Section S4) that discusses how our model can be used to capture other types of mutation, and reference this in Section 2.2. As part of this new section, we include an example of how to modify our cell equations for more targeted use-cases.

6. Having said that, I am surprised that the authors worked under the assumption that most mutations that remove the function of the synthetic construct affect regulation, as opposed to introducing a mutation that knocks out protein or transcription factor function (as the mutational space for such mutations is typically much larger). Exploring the differences between these types of mutations could be really interesting.

Response

Following from our answer to the reviewer's previous point, we wanted to keep our description of gene expression as simple as possible while allowing us to probe the effects of synthetic constructs design. As such, there is nothing special about our analysis of promoter mutations, just that it is one of the parts available from the host-aware cell equations that we use. Additionally, we often use promoter mutations in our analyses as they are convenient ways of disabling all downstream protein expression, analogous to a gene being 'fully inactivated'.

As we discussed above, our model is designed to strike a balance between complexity and usability. Users are free to consider more detailed the effects, such as the action of specific transcription factors, and they would simply have to modify the relevant cell equations. The framework's modularity makes this simple to do, and we provide an example of such equation modifications in our new section on "modelling other types of mutation" (Section S4).

7. However, the general approach, which disregards existing literature from the field of evolutionary biology and population genetics, prevents this work from having the broader appeal I think is needed for publication in a broad readership journal like Nature Communications. I would recommend targeting a journal more focused on synthetic biology, as the concerns I expressed above would be less relevant to that audience.

Response

We hope that the reformulation of our introduction alongside our responses to the points above will help change the reviewer's outlook here. In particular, we made an effort to explain why the work has broader appeal during the opening of our new introduction. Furthermore, we want to draw particular attention to the fact that recent publication trends in computational synthetic biology have shown that understanding evolution in synthetic systems is currently a hot and important topic. To this end, Nature Communications has recently published pieces on how varied genetic parts, mutation heterogeneity, and evolutionary landscapes could be developed and used to predict the mutation dynamics of synthetic devices (see for example: 'Diverse genetic error modes constrain large-scale bio-based production' by Rugbjerg et al., 2018, and 'Towards an engineering theory of evolution' by Castle et al., 2021). In many ways, our modular framework can be viewed as the first tangible model that (i) builds upon the discussions presented in these papers and (ii) answers their calls for such computational tools.

Reviewers' Comments:

Reviewer #1:

Remarks to the Author:

In the previous review, a key issue that was raised was the lack of experimental evidence that validates the utility of the authors' modeling framework. In response to the previous comments, the authors provided good reasons as to why these experiments cannot be easily performed, but the ways in and extent to which this modeling framework serves useful beyond generating hypotheses then remains unclear.

In this reviewer's opinion, the primary contributions of this work then appear to be as follows: it (1) proposes a class of plausible models for how evolution affects synthetic circuits, (2) analyzes the behavior of models in the class for several circuits via simulation, (3) thereby generates plausible and interesting hypotheses about the degree to which the toggle switch and repressilator are robust to mutations, and (4) suggests plausible mitigation strategies for unwanted evolution in synthetic circuits, which could be experimentally tested (but have not yet).

The main claims of the paper are as follows: as well as being a powerful exploratory tool for researchers, we show how our framework is able to provide new insights into industrial applications, such as how the DNA design of synthetic constructs impacts long-term protein yield and genetic shelf-life. Finally, we show how our framework can provide deeper insights into DNA design when applied to classic gene networks such as the toggle switch and the repressilator.

This reviewer's main point of constructive criticism is that in lieu of experimental validation, the authors' claims that the modeling framework provides insight for applications seems insufficiently substantiated. Although the models derived through this framework for the specific circuits under investigation are plausible, they have not been validated, so it seems premature to say they can provide useful insight into real world systems.

The modeling framework does, however, produce models which display interesting behaviors, helping to generate hypotheses that could be useful for these applications if true. Thus, it would be more clear to say the modeling framework can produce models which generate hypotheses about the effects of evolution in real-world circuits, which could in the future lead to important insights into strategies for assessing and mitigating the effects of evolution in synthetic circuits.

Additional points of constructive criticism:

The authors still use the term "predict" often in the text, especially in the introduction where they describe gaps in the field. There is insufficient evidence to suggest, however, that the authors are helping to fill this gap. This should be made clear or the authors should focus on other gaps in their introduction.

The word "predict" is also used in claims about the modeling framework itself: "Given that good fits can be obtained in some instances without describing intermediate mutations, it is evident that our model strikes a balance between the number of parameters and the ability to predict complex behaviours." The model strikes a balance between the number of parameters and the ability to fit complex behaviors, which is necessary but not sufficient for predicting complex behaviors. The authors note thereafter that "this is not equivalent to obtaining a complete predictive association between a system's genetic design and its evolutionary dynamics," which seems to be acknowledging the difference between fitting data and predicting it, but what this sentence means in the context of the previous one is not clear enough. It would be better to avoid making any claims which imply that the model can make good predictions, since this has not been validated.

Similarly, Figure 2 is titled "Predicting protein yield and viability of protein production," which may lead a reader to suspect that this modeling framework's predictive capacity has been tested. A clearer title would be "Modeling protein yield and viability of protein production."

Similarly, in the conclusion the authors state that "we demonstrated a number of core use-cases, from predicting protein yields and genetic shelf-lives in growing populations, to uncovering new design paradigms of multi-gene constructs." However, there is insufficient evidence that the

predictions of the models derived through this framework for this purpose have good predictive capacity (nor that these design paradigms are useful for real-world applications). These facts should be made clear or these claims should be removed.

In line with the previous comments, the title of section 6.2: "A mutation-aware approach uncovers key applications for biotechnology and cell engineering," again seems premature. It may be better to say "A mutation-aware modeling framework generates testable hypotheses relevant to biotechnology and cell engineering."

Reviewer #2:

Remarks to the Author:

The authors have satisfactorily addressed most of my comments. In response to Q4, they seem to consider repressilator and toggle triad synonymous, but these are different circuits both of three nodes each. While a repressilator is defined as A inhibits B that inhibits C that inhibits A, toggle triad is defined as A, B and C inhibiting each other.

Reviewer #3:

Remarks to the Author:

The authors have done an excellent job addressing my concerns. I am very happy with the manuscript and have no issues with it being published.

Reviewer #1

In the previous review, a key issue that was raised was the lack of experimental evidence that validates the utility of the authors' modeling framework. In response to the previous comments, the authors provided good reasons as to why these experiments cannot be easily performed, but the ways in and extent to which this modeling framework serves useful beyond generating hypotheses then remains unclear.

In this reviewer's opinion, the primary contributions of this work then appear to be as follows: it (1) proposes a class of plausible models for how evolution affects synthetic circuits, (2) analyzes the behavior of models in the class for several circuits via simulation, (3) thereby generates plausible and interesting hypotheses about the degree to which the toggle switch and repressilator are robust to mutations, and (4) suggests plausible mitigation strategies for unwanted evolution in synthetic circuits, which could be experimentally tested (but have not yet).

The main claims of the paper are as follows: as well as being a powerful exploratory tool for researchers, we show how our framework is able to provide new insights into industrial applications, such as how the DNA design of synthetic constructs impacts long-term protein yield and genetic shelf-life. Finally, we show how our framework can provide deeper insights into DNA design when applied to classic gene networks such as the toggle switch and the repressilator.

This reviewer's main point of constructive criticism is that in lieu of experimental validation, the authors' claims that the modeling framework provides insight for applications seems insufficiently substantiated. Although the models derived through this framework for the specific circuits under investigation are plausible, they have not been validated, so it seems premature to say they can provide useful insight into real world systems.

The modeling framework does, however, produce models which display interesting behaviors, helping to generate hypotheses that could be useful for these applications if true. Thus, it would be more clear to say the modeling framework can produce models which generate hypotheses about the effects of evolution in real-world circuits, which could in the future lead to important insights into strategies for assessing and mitigating the effects of evolution in synthetic circuits.

Additional points of constructive criticism:

The authors still use the term "predict" often in the text, especially in the introduction where they describe gaps in the field. There is insufficient evidence to suggest, however, that the authors are helping to fill this gap. This should be made clear or the authors should focus on other gaps in their introduction.

The word “predict” is also used in claims about the modeling framework itself: “Given that good fits can be obtained in some instances without describing intermediate mutations, it is evident that our model strikes a balance between the number of parameters and the ability to predict complex behaviours.” The model strikes a balance between the number of parameters and the ability to fit complex behaviors, which is necessary but not sufficient for predicting complex behaviors. The authors note thereafter that “this is not equivalent to obtaining a complete predictive association between a system’s genetic design and its evolutionary dynamics,” which seems to be acknowledging the difference between fitting data and predicting it, but what this sentence means in the context of the previous one is not clear enough. It would be better to avoid making any claims which imply that the model can make good predictions, since this has not been validated.

Similarly, Figure 2 is titled “Predicting protein yield and viability of protein production,” which may lead a reader to suspect that this modeling framework’s predictive capacity has been tested. A clearer title would be “Modeling protein yield and viability of protein production.”

Similarly, in the conclusion the authors state that “we demonstrated a number of core use-cases, from predicting protein yields and genetic shelf-lives in growing populations, to uncovering new design paradigms of multi-gene constructs.” However, there is insufficient evidence that the predictions of the models derived through this framework for this purpose have good predictive capacity (nor that these design paradigms are useful for real-world applications). These facts should be made clear or these claims should be removed.

In line with the previous comments, the title of section 6.2: “A mutation-aware approach uncovers key applications for biotechnology and cell engineering,” again seems premature. It may be better to say “A mutation-aware modeling framework generates testable hypotheses relevant to biotechnology and cell engineering.”

Response

In line with Reviewer 1’s comments, we have:

- Removed all suitable instances of the word “predict” throughout the manuscript, such that we are no longer making undue claims
- Carefully rephrased the end of the abstract and the final paragraph of the introduction to avoid making claims that our model “generates insights”, and rather that it generates hypotheses with a range of implications
- Changed the title of Section 6.2 to “Our framework brings forth hypotheses with broad applications”

We hope the reviewer agrees that these changes result in a fairer account of our contributions, and we thank them for guiding our usage of technical language throughout this process.

Reviewer #2

The authors have satisfactorily addressed most of my comments. In response to Q4, they seem to consider repressilator and toggle triad synonymous, but these are different circuits both of three nodes each. While a repressilator is defined as A inhibits B that inhibits C that inhibits A, toggle triad is defined as A, B and C inhibiting each other.

Response

We thank the reviewer for spotting this error. In revising our manuscript to conform to Nature Communications' editorial guidelines, we made the discussion more concise. In turn, we now make passing reference to the study of Harlapur et al., but do not discuss its findings in detail. As such, any reference to toggle triads is now gone. We hope that the reviewer finds this appropriate.

Reviewer #3

The authors have done an excellent job addressing my concerns. I am very happy with the manuscript and have no issues with it being published.

Response

We thank the reviewer for their comments, and overall for making our manuscript's content better suited to a wider audience.